# Optimizing Iodide-Adduct CIMS Quantitative Method for Toluene Oxidation Intermediates: Experimental Insights into Functional Group Differences

Mengdi Song[1,3], Shuyu He[1,3], Xin Li[1,2,3], Ying Liu[1,3], Shengrong Lou[4], Sihua Lu[1,3], Limin Zeng[1,3], and
Yuanhang Zhang[1,3]

[1]State Key Joint Laboratory of Environmental Simulation and Pollution Control, College of Environmental Sciences and Engineering, Peking University, Beijing, 100871, P.R. China
[2]Collaborative Innovation Center of Atmospheric Environment and Equipment Technology, Nanjing University of Information Science & Technology, Nanjing, 210044, P.R. China
[3]International Joint Laboratory for Regional Pollution Control, Ministry of Education, Beijing, 100816, P.R. China
[4]State Environmental Protection Key Laboratory of Formation and Prevention of Urban Air Pollution Complex, Shanghai Academy of Environmental Sciences, Shanghai, 200233, P.R. China

*Correspondence to*: Xin Li (li_xin@pku.edu.cn)

**Abstract.** Iodide-Adduct time-of-flight chemical ionization mass spectrometry (I-CIMS) has been developed as a powerful tool for detecting the oxidation products of volatile organic compounds. However, the accurate quantification of species that do not have generic standards remains a challenge for I-CIMS application. To accurately quantify aromatic hydrocarbon oxidation intermediates, both quantitative and semi-quantitative methods for I-CIMS were established for intermediate species. The direct quantitative experimental results reveal a correlation between sensitivity to iodide addition and the number of polar functional groups (keto groups, hydroxyl groups, and acid groups) present in the species. Leveraging the selectivity of I-CIMS for species with diverse functional groups, this study established semi-quantitative equations for four distinct categories: monophenols, monoacids, polyphenol or diacid species, and species with multiple functional groups. The proposed classification method offers a pathway to enhance the accuracy of the semi-quantitative approach, achieving an improvement in $R^2$ values from 0.50 to beyond 0.88. Overall, the categorized semi-quantitative method was utilized to quantify intermediates formed during the oxidation of toluene under both low NO and high NO conditions, revealing the differential variations in oxidation products with varying levels of NOx concentration.

## 1 Introduction

Volatile organic compounds (VOCs) react with oxidants (e.g., hydroxyl radical OH, ozone $O_3$, nitrate radical $NO_3$, etc.) in the atmosphere and contribute significantly to the formation of secondary organic aerosols (SOAs) (Hu et al., 2007). However, accurate prediction of SOAs remains a challenge, partly due to an insufficient understanding of SOA precursors which mainly consist of oxidation intermediates of VOCs (Bloss et al., 2005; Li et al., 2019a; Nehr et al., 2014; Ng et al., 2007). As one of

the most important VOCs in the urban atmosphere, aromatic hydrocarbons react with OH producing various intermediate products (Li et al., 2019b; Wu and Xie, 2017; Molteni et al., 2018; Schwantes et al., 2017; Wu et al., 2014), most of which are oxygenated volatile organic compounds (OVOCs). According to oxidation pathways, the intermediates can be divided into four categories (Song et al., 2021): (1) aldehyde pathway: aromatic aldehyde compounds such as benzaldehyde and methyl

benzaldehyde; (2) phenolic pathway: phenol, cresol, and polyhydroxy aromatic phenols, (3) bicyclic peroxy radical pathway: furanone, glyoxal, methyl glyoxal, and (4) epoxide pathway: epoxides. The intermediate oxidation products of these four pathways contain many different functional groups; therefore, the products are challenging to measure.

Chemical ionization mass spectrometry (CIMS) techniques allow classification based on exact molecular weight for nearly all semi-volatility and low volatility intermediate species (Bianchi et al., 2019; Riva et al., 2019a), but appropriate ion sources

need to be selected for different types of intermediate products. CIMS ion sources mainly include positively charged $H_3O^+$(Schwantes et al., 2017) and $NH_4^+$(Riva et al., 2019a), and negatively charged $NO_3^-$ (Xu et al., 2020), $I^-$ (Lee et al., 2014),$CH_3CH(O)O^-$ (Hansel et al., 2018), and $CF_3O^-$ (Schwantes et al., 2017), etc. Among them, $H_3O^+$ ions in conventional PTR instrument are designed to primarily measure VOCs (Yuan et al., 2017; Riva et al., 2019b). However, with the inlet modifications introduced in the newly developed VOCUS or FUSION PTR instruments, the $H_3O^+$ ions were able to measure

a large range of OVOCs (Reinecke et al., 2023; Riva et al., 2019b). Moreover, $NH_4^+$ ions are used for the detection of oxygenated organic compounds, including alcohols, aldehydes, ketones (Hansel et al., 2018; Xu et al., 2022), $NO_3^-$ ions are used for the detection of highly oxygenated organic molecules (HOMs) (Xu et al., 2020), $I^-$ ions are used for the detection of compounds containing many different functional groups, including monophenols, polyphenols, monoacids, diacids, phenolic acids, keto acids, and inorganic species (Lee et al., 2014), $CH_3CH(O)O^-$ ions are used for the detection of organic acids (Hansel

et al., 2018), and $CF_3O^-$ ions are used for the detection of oxygenated organics, including hydroperoxides (Schwantes et al., 2017). I-CIMS has often been used in field observations and laboratory research in recent years (Lee et al., 2014; Ye et al., 2021; Isaacman-Vanwertz et al., 2018; Zhang and Zhang, 2021; Coggon et al., 2019; Wang et al., 2020) because it induces minimal fragmentation (Lee et al., 2014) and shows a good detection capacity for $HNO_3$, $N_2O_5$, halogenated organic matter, and OVOCs containing carboxyl, epoxide, and multifunctional groups (Iyer et al., 2016; Dorich et al., 2021; Veres et al., 2015).

Although numerous I-CIMS applications exist for investigating VOC atmospheric chemistry, challenges associated with the qualitative / quantitative detection of critical reaction products persist (Riva et al., 2019a). First, the quantification of oxidation intermediates requires calibration using commercial standards. Direct calibration plays a crucial role in exploring the sensitivity characteristics of instruments and reducing the uncertainty of quantitative methods. Common direct calibration methods include the utilization of standard gas cylinders (SGC), penetrant tubes (CPT), and liquid calibration units (LCU) (Xu et al.,

2022; Huang et al., 2019). Xu et al. employed 60 organic compound standards utilizing SGC and a home-built LCU method to calibrate $NH_4^+$ CIMS, revealing its differential sensitivity to diverse organic compounds (Xu et al., 2022). Additionally, Li et al. implemented 22 organic standards with the LCU method for I-CIMS calibration, achieving a significant reduction in total organic carbon concentration uncertainty to approximately 20%-35% when coupled with the voltage scanning approach (Li et al., 2021). However, most intermediates measurable by I-CIMS are difficult to synthesize effectively as pure standards. Second,

during the direct calibration process, because most of standard samples belong to semi-volatility species, they are present a liquid or solid state, which are difficult to calibrate. Some studies use Filter Inlet for Gases and AEROsols (FIGAERO) to calibrate species with very low volatility (Ye et al., 2021; Lopez-Hilfiker et al., 2014), but this process presents challenges with controlling the calibrated humidity status. In a semi-quantitative process, lyer et al. discovered the linear relationship between the cluster binding enthalpies and logarithmic instrument sensitivities (Iyer et al., 2016). Then, Lopez-Hilfiker et al.

discovered that ionization declustering analysis of iodine additions could be performed by changing instrument voltage (Lopez-Hilfiker et al., 2016). The above two semi-quantitative methods have, for the first time, achieved an approximate sensitivity analysis for species that cannot be calibrated using traditional standard calibration methods. However, the wide range of sensitivities of an iodide CIMS to different species poses a significant challenge to accurately assess the sensitivity of species with different functional groups(Bi et al., 2021a). Compared to Proton-Transfer-Reaction Time-of-Flight Mass Spectrometry

(PTR-ToF-MS), which exhibits sensitivity variations of only 0.3-0.4 orders of magnitude when measuring OVOCs (Bi et al., 2021a; Sekimoto et al., 2017), I-CIMS shows sensitivity changes of 4-5 orders of magnitude (Lee et al., 2014; Ye et al., 2021). The sensitivity of I-CIMS varies significantly across different functional group types of species, indicating that a single semi-quantitative equation is insufficient for quantifying all detected species with this technique. Additionally, the complex theoretical calculation methods underlying binding energy calculations, as well as the requirement for stable product

concentrations in voltage scanning, impose limitations on semi-quantitative methods. Moreover, the presence of isomers affects all mass spectrometry quantification. This results in an uncertainty in the sensitivity obtained by these methods could reach 0.5-1 order of magnitude for a single compound (Bi et al., 2021c). Even when estimating the uncertainty in measuring total oxidation product concentrations, it reaches about 60% (Isaacman-Vanwertz et al., 2018). To obtain values closer to the actual sensitivity, this semi-quantitative process needs to be improved (Isaacman-Vanwertz et al., 2018; Heinritzi et al., 2016).

In this study, an in-depth exploration of quantitative and semi-quantitative methods within the I-CIMS measurement process was undertaken to enhance the identification and quantification of intermediates in the oxidation process of aromatic hydrocarbons. Direct quantitative calibration was conducted with 37 OVOC species associated with the oxidation products of aromatic hydrocarbons. Based on the obtained direct quantitative sensitivity data set, we discussed the corresponding differences in the sensitivity of I-CIMS to different functional groups OVOCs. The semi-quantitative equation was established

using theoretical computational methods. In order to enhance the accuracy of the semi-quantitative approach for species, a classification method was applied to species with different functional groups. In addition, the quantitative and semi-quantitative methods were applied to the experimental study on the oxidation of aromatic hydrocarbons in the chamber experiements, which ensured the feasibility of this method. Additionally, this method quantified the challenging-to-quantify ring-retaining and ring-opening products formed during the oxidation of toluene, and discussed their differences in proportions

under different NOx conditions.

## 2 Experimental Description

### 2.1 CIMS measurement

The iodide-adduct high-resolution time-of-flight chemical ionization mass spectrometry (I-CIMS) used in this study was a commercial product from TOFWERK. The sample flow was approximately 2 L/min, comparable to the primary ion flow rate. The primary ion $I^-$ was generated by introducing 3 ml/min 300 ppm methyl iodine standard gas in 2.3 L/min pure nitrogen through an X-ray source in an ion molecule reaction (IMR) chamber. The volume of the IMR chamber was 47 cm$^3$ and the working pressure was maintained at approximately 380 mbar. During measurements, the variation in IMR pressure is controlled within ±3 mbar. In the IMR chamber, neutral molecules (X) reacted with iodine reagent ions ($I^-$) to produce different product ions ($XI^-$). The main reaction path is shown in R1:

$$X + I^- \rightarrow XI^- \tag{R1}$$

In the IMR, the sample and primary ions were mixed and interacted for approximately 120 milliseconds. Then, the mixed flow was passed through an orifice into the high-resolution time-of-flight mass spectrometry and arrived at the detector to be classified by exact molecular weight. During the experimental operating conditions of this study, the iodide CIMS exhibited a total ion count (TIC) of approximately 2 ions per extraction (ions/ex.) and 32,000 counts per second (cps). The resolving power of the I-CIMS was 5300–5600. The data acquisition frequency of the I-CIMS instrument was 1 s.

TofWare software version 3.2.2 (Tofwerk Inc.) was used for high-resolution peak fitting of I-CIMS data. For mass spectrometry analysis, we used the single-ion peaks for $I^-$, $H_2OI^-$, $HNO_3I^-$, and $I_3^-$ for mass calibration, and ensured that the absolute in-flight deviation of the m/Q was below 5 ppm (2σ), which was much lower than the instrument guideline of 20 ppm. Then, the molecular ion peak was standardized to obtain the final signal data. The signal standardization methods are obtained using Equation (1) (Lee et al., 2014; Ye et al., 2021).

$$\text{Normalized Signal (ncps)} = \frac{\text{Signal}}{I^- + H_2OI^-} * 10^6 \tag{1}$$

### 2.2 Calibration method

In this study, 37 species with different functional group were directly calibrated using certified penetrant tubes (CPT) and a home-built liquid calibration unit (LCU), including monophenols, monoacids, polyphenols, diacids, phenolic acids, keto acids, furanones, and other species (Table S1).

For OVOCs that can be customized to a standard gas and penetrant tubes, calibration is often performed using certified penetrant tubes (KinTek Inc.) at 5-6 gradient concentration levels (Huang et al., 2019). The calibrated concentration ranges from dozens of ppt to several ppb levels. However, because many standard samples are liquid or solid, it is challenging to make permeable tubes that have stable permeability. Because this study focused on the gas phase reaction, an appropriate home-built liquid calibration unit was designed so the OVOCs could be calibrated in the gasous form under normal temperature and pressure. The standard sample was mixed with a soluble solvent, including water, dichloromethane, or acetone, and the

solvent was atomized at a given flow rate. Subsequently, the atomized gas was mixed with high-flow nitrogen to ensure the complete evaporation of the atomized droplets, which were then injected into the sampling port. No liquid condensation was observed on the wall of the mixing unit, and no particulate matter was present. After sufficient equilibration time, stable signals

of standard samples could be detected in I-CIMS. The specific calibration method can be found in a study by Qiu et al. (Qiu et al., 2021) and Qu et al. (Qu et al., 2023).

To investigate the influence of humidity on calibration, both CPT and LCU calibration system are equipped with a humidification section that can control humidity within the range of 0.12 to 22.00 mmol/mol, which corresponds to a relative humidity (RH) of 0.4% to 70% at a temperature of 25°C and a pressure of 101.325 kPa. During the calibration process, by

adjusting the humidity, the changing relationship between the sensitivities of various standard samples and the water vapor pressure can be obtained.

In this study, the molecular weight range of directly calibrated species was 46.01 to 216.17, which covered the molecular weight range of the principal gaseous intermediates of toluene and o-cresol (48.04-203.15). The linear correlation between the normalized signal values of the directly calibrated species and the concentration was excellent, with $R^2$ values greater than

0.99 for most species. For species whose sensitivities could be directly calibrated, concentrations can be calculated using Equation (2):

$$[X\_ppb] = \frac{\text{Normalized signal}}{\text{Sensitivity from direct quantification} \times \text{RH}_{Corr}} \quad (2)$$

where $\text{RH}_{Corr}$ represents the humidity correction equation. The humidity correction equation, detailing the changing relationship between sensitivities of various standard samples and water vapor pressure, is presented in Section S1 and visually

illustrated in Figure S2 of the Supplement. Additionally, the humidity correction for species without standard samples can be estimated based on the characteristics of species with similar functional groups.

Assuming that the random uncertainty of the CIMS detector counts follows Poisson statistics, the signal-to-noise ratio (S/N) and the detection limits for calibrated species can be calculated using Equation (3) (Bertram et al., 2011):

$$\frac{S}{N} = \frac{C_f[X]t}{\sqrt{C_f[X]t + 2Bt}} \quad (3)$$

Where [X] represents the detection limits (ppbv), $C_f$ is the sensitivity factor from calibration (ncps s$^{-1}$ ppbv$^{-1}$), t is the integration time (s), B represents the background normalized signal rate (ncps s$^{-1}$). In this study, we calculate the detection limits for all 37 calibrated species under 1 second averaging and a signal-to-noise ratio of 3. The data results are listed in Table S1.

**2.3 Binding Energy Calculation Method**

The semi quantitative method of I-CIMS can be established by two approaches. The first approach, referred to as the

binding enthalpy method, utilizes the binding energy between a given species and iodide ions to construct a parametric equation for the sensitivity. This method is extensively discussed and implemented in this study. The second approach, known as the scanning voltage method, relies on leveraging the instrument's transmission efficiency of molecular ions through

the electric fields to establish the parameterized equation of sensitivity and the mass-to-charge ratio. The details of the scanning voltage method are shown in Section S2. The binding energy method for the I-CIMS instrument requires four factors to be considered, i.e., the normalization of the instrument signal, binding characteristics of species and reagent ions,(Iyer et al., 2016) mass transmission correction (Isaacman-Vanwertz et al., 2018; Heinritzi et al., 2016) and humidity correction (Ye et al., 2021; Lee et al., 2014). These four factors are combined to generate a detailed semi-quantitative expression, as shown in Equation (4):

$$[\text{X\_ppb}] = \frac{\text{Normalized signal}}{\text{Sensitivity from binding enthalpy} \times \text{MassTrans} \times \text{RH}_{\text{Corr}}} \tag{4}$$

where MassTrans represents the mass transmission correction equation. The mass transmission correction equation characterizes the ability of the mass spectrometer to introduce ions with different mass-to-charge ratios from the IMR to the mass detector (Heinritzi et al., 2016). A detailed information can be found in Section 3.2.

In this study, theoretical and computational approach were used to calculate the binding energy between the species and iodide anion. The study demonstrates that iodide anion binds with species through hydrogen bonding(Zhang et al., 2020). The primary types of hydrogen bonds that can bind with iodide anion are N−H, O−H, and C−H. The type, quantity, and various hydrogen bond binding geometries in species all influence the magnitude of binding energy. The theoretical and computational calculations were carried out using the Gaussian 16 package(Gaussian 16, 2016). The standard species and oxidation intermediates were optimized at the B3LYP/6-31G* (Petersson et al., 1988; Petersson and Al-Laham, 1991) level of theory. In density functional theory (DFT), the PBE(Perdrew et al., 1997) and B3LYP methods have been reported to perform well in the calculation of iodide anion binding energies(Zhang et al., 2020; Iyer et al., 2016). In the calculation of binding energies between iodide ions and their respective standard species and oxidation intermediates, the London-dispersion effects cannot be neglected. Since B3LYP and PBE lack a description of dispersion corrections and significant improvements are achieved with the addition of D3 dispersion corrections(Goerigk et al., 2017), this study incorporated DFT-D3 for correction in the theoretical calculation process.

Therefore, all geometrical optimization of standard species, products, and iodide ions was performed using these theoretical methods: PBE/SDD (Schaefer, 2013), PBE/SDD-D3, and B3LYP/Def2TZVP-D3 (Weigend and Ahlrichs, 2005; Weigend, 2006) levels. Vibrational frequency analyses were employed on the optimized structures to verify their representation as true energy minima on the potential energy surface, ensuring the absence of any imaginary frequency. The binding energy of the product ion ($XI^-$) can be defined by subtracting the computed electronic energies of iodide ions ($I^-$) and reference species ($X$) from $XI^-$, and taking the absolute value (Iyer et al., 2016).

### 2.4 Chamber Experiments

To test whether the oxidation intermediates of aromatic hydrocarbons could be quantitatively identified using I-CIMS, oxidation experiments were carried out in a chamber for toluene. The chamber consisted of a Teflon FEP bag with a cylindrical

shape. The chamber was 3.2 meters long and 2 meters in diameter, with a practical volume of 10 $m^3$. It was equipped with 60
black light sources (330-400 nm) to simulate solar radiation. Nitrogen dioxide ($NO_2$) was used to characterize the photolysis performance of the chamber. The photolysis rate of $NO_2$ was estimated by a steady-state chemical actinometry method (Zou et al., 2016). The photolysis rate of $NO_2$ ($jNO_2$) was $2.59 \times 10^{-3}$ $s^{-1}$ when the black light tube was fully operated. The experimental chamber layout is shown in (Figure S3). Before the experiment, the chamber was cleaned with 100 L/min dry synthetic air (made from liquid $N_2$ and $O_2$ with a ratio of 80:20, purity > 99.999%) for at least 10 hours. The relative humidity of the chamber was humidified to approximately $55 \pm 5\%$, while the temperature was maintained at $26 \pm 1$ ℃. Following the comprehensive purification process, the residual concentrations of NOx within the chamber were notably low, specifically measuring under 0.4 ppb, nearing the detection thresholds of the commercial chemiluminescence technology instrument (Thermo Scientific™ Model 42i). Due to the inevitable presence of background NOx concentrations, the heterogeneous reaction of $NO_2$ on the Teflon surface within the chamber results in the release of nitrous acid (HONO) (Chu et al., 2022; Rohrer et al., 2005; Li et al., 2019a). Then, VOC precursor and NO were introduced into the chamber, leading to initial toluene concentrations of 100 ppbv (without NO injection) and 80 ppbv (with 60 ppb NO) in the chamber. The initial NO/toluene ratios were 0.01 and 0.75, respectively, corresponding to the low NO and high NO conditions in this study. After the chamber air became stable (within 10-20 min), the lights were turned on. Subsequently, upon illumination, the OH radical generated through the photolysis of HONO serves as a crucial source of free radicals. The photolysis of HONO led to the formation of OH radicals ranging from $1.23 \times 10^6$ molecule $cm^{-3}$ to $3.55 \times 10^6$ molecule $cm^{-3}$, which triggered the atmospheric oxidation reaction of aromatic hydrocarbons. During the chamber experiment, we often introduced synthetic air to compensate for the sampling instrument to keep the chamber volume and pressure constant (Novelli et al., 2020; Poppe et al., 2007). Usually, stable and reaction-independent tracers such as acetonitrile ($C_2H_3N$), hexafluorobenzene ($C_6F_6$), and sulfur hexafluoride ($SF_6$) are utilized to correct for dilution in the chamber.(Chu et al., 2022) In this study, we choses acetonitrile as the dilution tracer. During the chamber experiments, I-CIMS was employed for measuring intermediate oxidation products of aromatic hydrocarbons. For detailed information on the measurement methods for precursors, NOx, $O_3$, and temperature-humidity, refer to Section S3.

# 3 Result and discussion

## 3.1 Sensitivity of typical toluene oxidation intermediates in I-CIMS

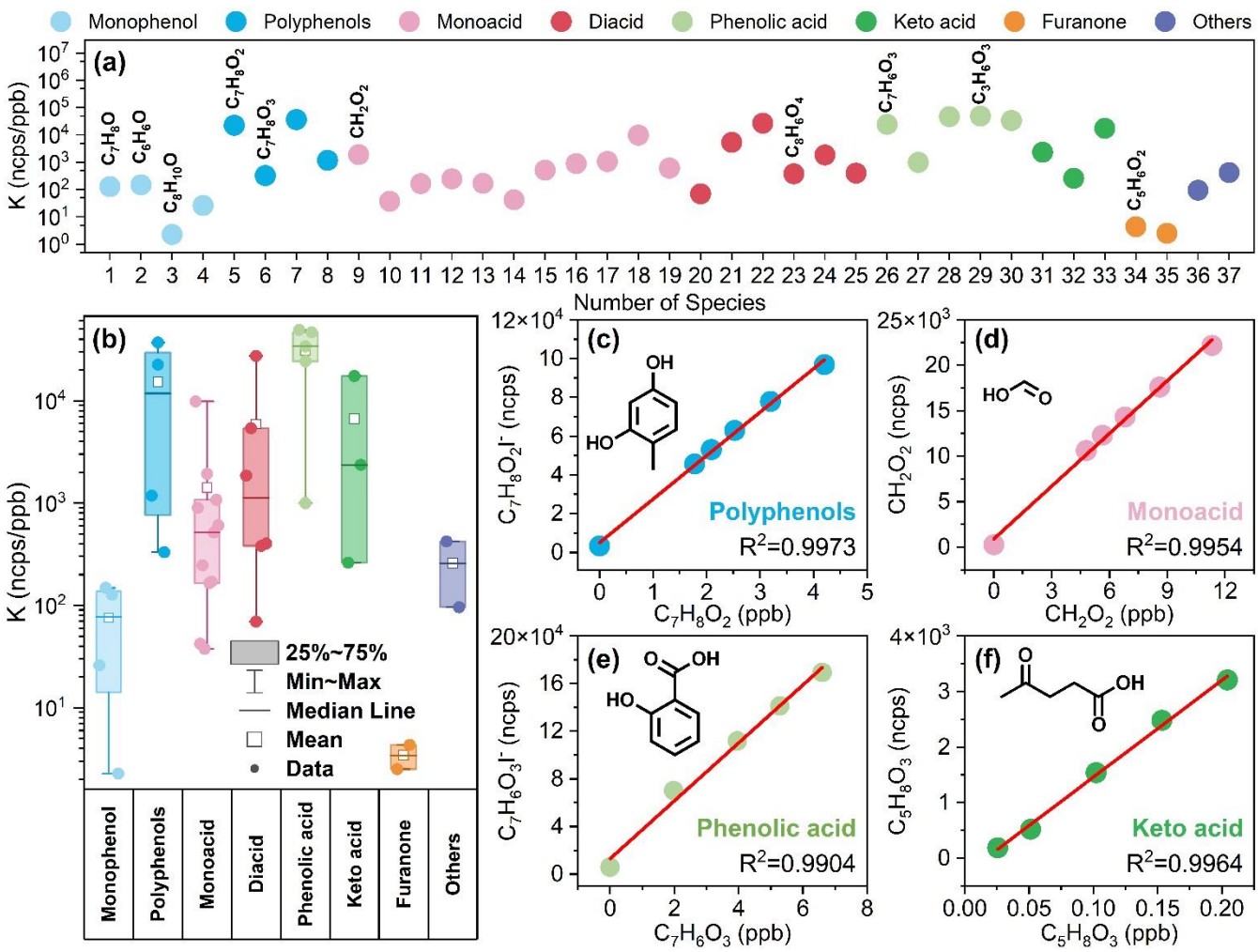

Figure 1: (a) Direct quantitation sensitivity results of 37 standard materials (b) Sensitivity statistics for standard materials containing different functional groups (c)-(f) Calibration curves of 2, 4-dihydroxytoluene ($C_7H_8O_2$), formic acid ($CH_2O_2$), salicylic acid ($C_7H_6O_3$), and levulinic acid ($C_5H_8O_3$). Note. Details of species with corresponding serial numbers in figure (a) are available in TableS1. All sensitivity values presented in the figures were acquired under the RH condition of approximately $55 \pm 5\%$.

For toluene oxidation intermediates with standards, such as m-cresol ($C_7H_8O$), 2,4-dihydroxytoluene ($C_7H_8O_2$), and 2,4,6-trihydroxytoluene ($C_7H_8O_3$), the directly calibrated sensitivity in I-CIMS is $1.3 \times 10^2$ ncps/ppb, $2.2 \times 10^4$ ncps/ppb, and $3.3 \times 10^2$ ncps/ppb, respectively (Figure 1 and Table S1). In the case of toluene oxidation intermediates lacking standards, this study selects standard samples with similar reactive functional groups to toluene oxidation intermediates for calibration, so that the subsequent quantitative and semi-quantitative equations are more applicable to the toluene oxidation system. Typical oxidation

products of toluene include aromatic phenols, ring-retaining phenols, ring-opening acids, ring-opening keto acids, ring-opening phenolic acids, and ring-opening furanones, among others (He et al., 2023). Based on the characteristics of the aforementioned toluene oxidation intermediates, this study selected 37 standard samples for calibration, including the main types of monophenols, polyphenols, monoacids, diacids, phenolic acids, keto acids, and furanones, as shown in Figure 1 and Table S1. By quantifying the sensitivity of the iodine adduction of monophenols, polyphenols, monoacids, diacids, phenolic acids, keto acids, furanones, and other species in I-CIMS (Figure 1), it was shown that iodide ions have different sensitivities for species with different functional groups due to differences in binding energies (Iyer et al., 2016). The sensitivity difference for these species ranged from $10^0$-$10^4$ ncps/ppb, spanning over 4 orders of magnitude. Based on a comparison with previous studies (Ye et al., 2021; Lee et al., 2014), this study identified distinct sensitivity characteristics of I-CIMS towards various functional groups. As illustrated in Figure 1 and Figure S4, the sensitivity to iodine addition is correlated with the type and quantity of polar functional groups present in the species, including keto groups, hydroxyl groups, and acid groups.

As depicted in Figure 1b, the sensitivity of furanones, monophenols, and monoacids gradually increases, indicating that species with a single active group, such as keto, hydroxyl, and acid groups, exhibit increasing sensitivity in the order listed (Figure 1b). Among them, furanone containing keto groups were the least sensitive, and the sensitivities of furfural and 3-methyl-2(5H)-furanone were 3 ncps/ppb and 4 ncps/ppb, respectively. The detection limits for furfural and 3-methyl-2(5H)-furanone are also very high (Table S1), indicating that I-CIMS does not have an advantage in measuring furanones. The sensitivities of monophenolic compounds such as phenol and m-cresol were $1.5 \times 10^2$ ncps/ppb and $1.3 \times 10^2$ ncps/ppb, respectively. I-CIMS demonstrates good detection capability for phenol and m-cresol, with low detection limits of 0.11 and 0.08 ppb (in 1-second, S/N=3), respectively. However, it exhibits relatively lower sensitivity for larger mass compounds such as 2,6-xylenol and texanol, resulting in higher detection limits. Previous studies have also shown that I-CIMS has good sensitivity toward compounds containing carboxylic acid groups (Mcneill et al., 2007; Le Breton et al., 2012; Lee et al., 2014). Similarly, here we found that the sensitivity of monoacids was higher, and the sensitivities of formic acid, allylacetic acid, and 2-ethylhexanoic acid were $1.9 \times 10^3$ ncps/ppb, $1.1 \times 10^3$ ncps/ppb, and $8.9 \times 10^2$ ncps/ppb, respectively. I-CIMS exhibits low detection limits for monoacids, ranging from a few to 400 ppt (Table S1).

For species containing more than one hydroxyl group, sensitivity also increased with the addition of ketone, hydroxyl, and carboxyl groups in the order listed (Figure 1 and Figure S4). Keto acids, with a keto group and an acid group, show significantly lower sensitivity compared to phenolic acids, which feature a hydroxyl group and an acid group. The sensitivities of pyruvic acid, citric acid, salicylic acid, and glycolic acid were $2.6 \times 10^2$ ncps/ppb, $9.9 \times 10^2$ ncps/ppb, $2.4 \times 10^4$ ncps/ppb and $4.7 \times 10^4$ ncps/ppb, respectively. Moreover, the sensitivity toward dicarboxylic compounds was higher than that toward monocarboxylic compounds due to the increase in the number of active groups (Figure S4). The sensitivities of oxalic acid, adipic acid, and glutaric acid were $5.4 \times 10^3$ ncps/ppb, $2.7 \times 10^4$ ncps/ppb, and $1.9 \times 10^3$ ncps/ppb, respectively. Species with multiple reactive functional groups exhibit high sensitivity and low detection limits, with detection limits ranging from a few ppt to 150 ppt (Table S1), demonstrating the excellent detection capability of I-CIMS for these species.

Humidity has a significant influence on the sensitivity of iodine adducts (Ye et al., 2021; Lee et al., 2014). Through the establishment of humidity-dependent parametric equations, species sensitivity under different humidity conditions can be obtained. As elaborated in Section S1, this study established humidity-dependent parametric equations for four categories of compounds (Figure S2): (1) single active functional group compounds like acrylic acid, which show rapid sensitivity decline with increasing humidity, (2) multiple active functional group compounds like pinonic acid, which have higher sensitivity and are less affected by humidity, (3) polyphenol compounds like 2,4,6-trihydroxytoluene, which are nearly unaffected by humidity, and (4) small-molecular-weight acids like formic acid, which show increased sensitivity at low humidity but decreased sensitivity at higher humidity. These humidity-dependent parametric equations correspond to $RH_{Corr}$ in Equation (2), (4), and (S2).

## 3.2 Establishment of classification-based semi-quantitative equations

For toluene oxidation intermediates lacking standards, semi-quantitative methods are utilized to estimate their sensitivity in I-CIMS. When establishing the semi-quantitative method based on experimental sensitivity in the toluene system, consideration must be given to the four factors mentioned in Section 2.3. Alongside the previously discussed signal normalization and humidity correction, attention must be paid to mass transmission correction and the binding characteristics of species and iodide ions.

For the toluene oxidation system under investigation, this study analyzed the mass transmission effects of species within the mass range (180-350 m/z), where the primary gaseous oxidation products of toluene are located (Figure S5). It is shown that within the specified mass-to-charge ratio range, the mass discrimination effect has minimal influence on the sensitivity of the target species. This impact remains similarly negligible when comparing with mass transmission curves from prior studies (Ye et al., 2021) in the 180-350 m/z range. Therefore, when quantifying toluene oxidation products, the correction factor (MassTrans) of Equation (4) in the semi-quantitative process can be set to 1. However, for species with higher mass-to-charge ratios, it is crucial to account for mass correction using the mass transmission curves reported by Heinritzi et.al (Heinritzi et al., 2016) and Ye et. al (Ye et al., 2021).

When considering the binding characteristics of species and reagent ions, this study utilizes a reasonable linear relationship between the cluster binding enthalpy and logarithmic instrument sensitivity of iodine adducts (Iyer et al., 2016). This suggests that a relatively straightforward method could be used to predict the sensitivity of I-CIMS toward species with calculated binding energies. While this method is relatively simple, existing research has primarily focused on acidic species(Iyer et al., 2016). This study computed the binding energies of various functional group types, including monophenols, polyphenols, monoacids, diacids, phenolic acids, keto acids, etc. However, when the logarithmic relationship was directly fitted, the correlation was poor. The correlation between experimentally observed sensitivities and calculated binding enthalpies at the PBE/SDD level did not exhibit a strong relationship ($R^2$=0.34). Upon correction for D3 effects at B3LYP/Def2TZVP level, the performance was enhanced, resulting in an improved $R^2$ of 0.52 (Figure S6). However, the

correlation remained relatively weak, introducing a large degree of uncertainty to the subsequent calculations of intermediate product yield. For example, when examining the phenolic pathway for toluene, the yield of cresol obtained through semi-quantitative analysis was 2.5 times lower than that obtained through direct calibration.. The poorer correlation may be attributed to significant differences in the sensitivity of iodine ions to different functional group species. Earlier studies have primarily focused on acidic species, showing favourable correlations.

Given the significant selectivity distinctions of I-CIMS towards various reactive functional groups, the measured species were classified into four groups: monophenols, monoacids, polyphenol or diacid species, and species with multiple functional groups for semi-quantitative analysis (Figure 2 and Figure S6). It was observed that the categorization of these species enhances the accuracy of logarithmic fitting between binding energy and sensitivity. Higher theoretical level calculations and the inclusion of dispersion corrections enhance the quality of logarithmic fitting between binding energy and species sensitivities, as depicted in Figure S6. The B3LYP/Def2TZVP (D3) level of calculation is computationally less expensive compared to DLPNO-CCSD(T) method (Iyer et al., 2016) and has shown favourable performance in describing the relationship between binding energy and experimental sensitivity of species. Therefore, this theoretical calculation level was chosen to establish a semi-quantitative equation for binding enthalpies and sensitivity.

For monophenol species, with a single hydroxyl group, the most energetically favourable cluster geometry involves the attachment of the iodide ion to the hydroxyl group H via a single hydrogen bond (Figure 2a). These species include phenol, m-cresol, 2,6-xylenol, and texanol, with binding enthalpies to iodide ion between 16 to 19 kcal/mol. A strong logarithmic linear correlation is observed between binding enthalpies and sensitivity, with $R^2$ values of 0.92 (Figure 2a). These species exhibit relatively low binding enthalpies and sensitivities. The slope of the fitting between binding enthalpies and sensitivities is higher compared to other categories, indicating a significant impact of binding energy variations on sensitivity.

For monoacid species, with a single acid group, the most energetically favourable cluster geometry involves the attachment of the iodide ion to the acid group H via a single hydrogen bond (Figure 2b).These species, such as formic acid, acrylic acid, and propionic acid, have binding enthalpies to iodide ions ranging from 17 to 28 kcal/mol. It shown a good logarithmic linear correlation exists between binding enthalpies and sensitivity, yielding $R^2$ values of 0.90 (Figure 2b). The sensitivity and binding energy to iodide of these species are significantly higher than those of monophenol species.

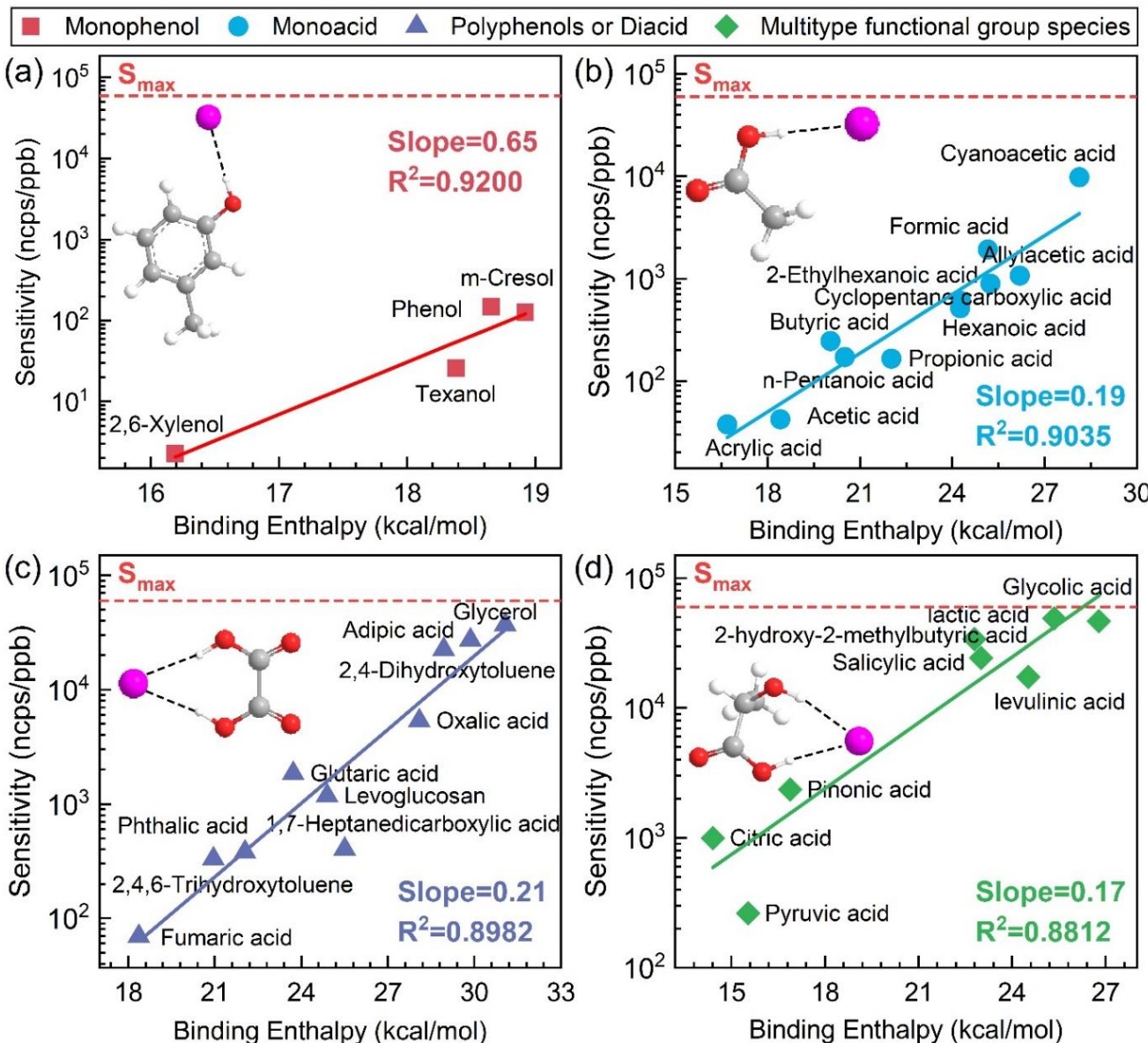

**Figure 2: Fitting curve for cluster binding enthalpies and logarithmic sensitivities of (a) monophenol species (b) monoacid species, (c) polyphenol or diacid species, and (d) multitype functional group species at the B3LYP/Def2TZVP (D3) level. All sensitivity values presented in the figures were acquired under the RH condition of approximately 55 ± 5%.**

Polyphenol or diacid species, compounds with multiple carboxyl and acid groups, exhibit a tendency for iodide-adduct to form two hydrogen bonds, as illustrated in Figure 2c. These species, such as fumaric acid, phthalic acid, and 2,4-dihydroxytoluene, with binding enthalpies to iodide ion between 19 to 31 kcal/mol. The fitting performance ($R^2$) for the correlation between

binding enthalpies and sensitivity is 0.90 (Figure 2c). These species, characterized by identical reactive functional groups, generally form two symmetric hydrogen bonds, resulting in higher binding energy and sensitivity.

For species with multiple functional groups, primarily consisting of phenolic acids and keto acids, examples include salicylic acid, citric acid, and lactic acid, where iodide-adducts tend to form two different hydrogen bonds (Figure 2d). The fitting performance ($R^2$) for the correlation between binding enthalpies and sensitivity is 0.88 (Figure 3d). The sensitivity of these compounds is significantly higher than that of other categories, with binding enthalpies to iodide ion ranging between 15 to 27 kcal/mol. Among them, citric acid is quite unique, as it has three carboxyl groups and one hydroxyl group. When reclassified as a diacid or polyphenol species, the $R^2$ weakened to 0.62, and the relative deviations between measured sensitivity and calculated sensitivity increased more than twofold, reaching 88%. This indicates that species containing even one different type of functional group should preferably be classified into the multiple functional groups category.

When using semi-quantitative equations to calculate species sensitivity, the maximum sensitivity ($S_{max}$) is considered a reasonable upper bound limit. The maximum sensitivity of I-CIMS was empirically found to be 19-22 cps/ppt under the operating conditions of the instruments employed by Lee, Iyer, and Lopez-Hilfiker et al. (Lee et al., 2014; Iyer et al., 2016; Lopez-Hilfiker et al., 2016). The maximum sensitivity (60 ncps/ppt) selected in this study refers to the findings reported by Ye et al. under similar IMR working conditions. In Figure 2, we use red dashed lines to indicate the maximum sensitivity of the I-CIMS instrument. Similarly, I-CIMS exhibits a minimum sensitivity. When the binding energy between the analyte species and I- approaches or falls below the binding energy of $H_2OI^-$ at 12 kcal/mol at the B3LYP/Def2TZVP (D3) level, the instrument cannot detect the species.

By applying a classification-based approach to establish semi-quantitative equations according to different functional groups, this study significantly improved the fitting effect of the equations, raising the $R^2$ from 0.50 to above 0.88 and reducing the uncertainty of the semi-quantitative method. This classification method also applies to previous research findings (Figure S7), enhancing the accuracy of quantification for both monoacid and multiple functional group species, with $R^2$ increasing from 0.66 to above 0.90. Particularly noteworthy is the more than fivefold improvement in the semi-quantitative coefficient for monoacid species (Figure S7). Therefore, classifying semi-quantitatively based on the selective differences of I-CIMS among different species is crucial for improving the accuracy of semi-quantitative methods.

The semi-quantitative uncertainty was computed by dividing the absolute difference between the measured sensitivity and the calculated sensitivity by the measured sensitivity, as illustrated in Figure 3. For standard samples, the findings indicated a satisfactory concordance between the calculated and experimental sensitivity factors, with relative deviations below 40% (Figure 3a). As can be observed from Figure 3b, the classified semi-quantitative method based on the binding energies (C-SS in Figure 3b) enhances the accuracy of quantification.

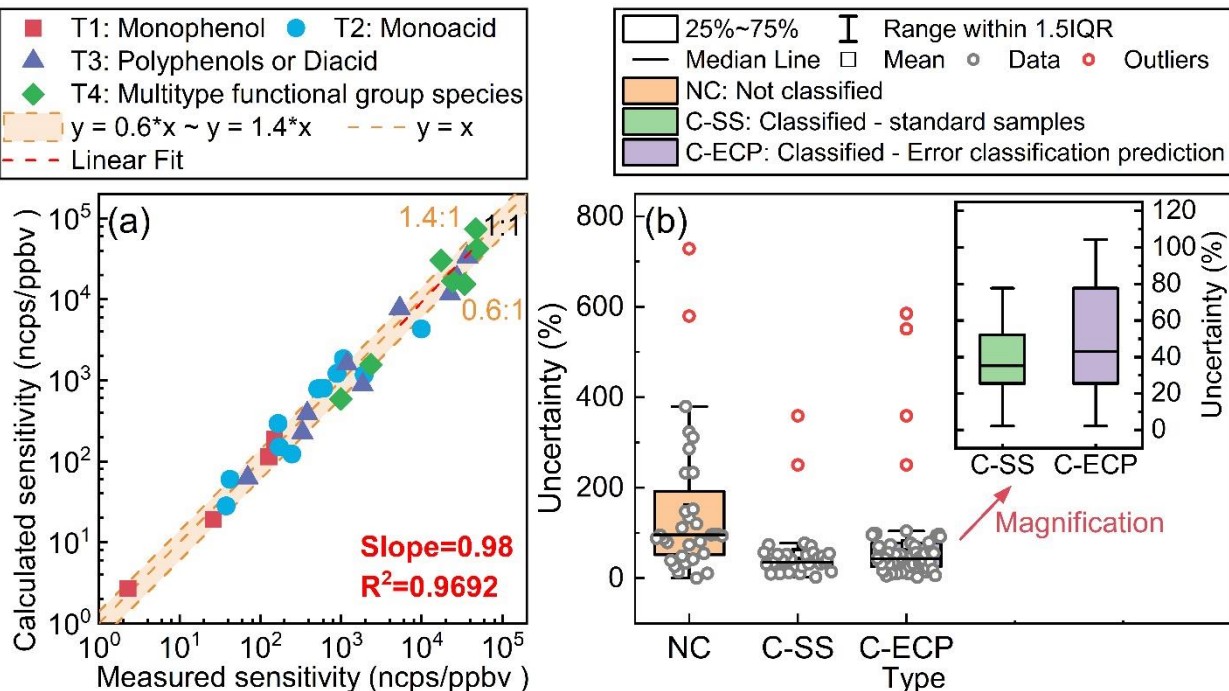

**Figure 3: (a) The difference between the measured sensitivity and the calculated sensitivity for standards at the B3LYP/Def2TZVP (D3) level. (b) The uncertainty of classification-based semi-quantitative methods at the B3LYP/Def2TZVP (D3) level. The uncertainty is computed by dividing the absolute difference between the measured sensitivity and the calculated sensitivity by the measured sensitivity. All sensitivity values presented in the figures were acquired under the RH condition of approximately $55 \pm 5\%$.**

However, for species without standard samples, the selection of classification becomes paramount, as inappropriate classification may introduce significant uncertainties. In this research, since the classification features of monophenols and monoacids are quite distinct, we can disregard the errors caused by misclassification. For the third and fourth groups of species, a more detailed classification is possible, distinguishing them into polyphenol, diacid, phenolic acid, and keto acid species. This refined classification leads to new semi-quantitative relationships, as illustrated in Figure S8. Taking the fourth category of multitype functional group species as an example, when we assume that it contains only phenolic acid species and erroneously apply this equation to quantify keto acid species, it will introduce significant uncertainties due to misclassification. Furthermore, when the multitype functional group species contain two acid groups and one hydroxyl group, the uncertainty arising from their misclassification as diacid species also needs to be taken into consideration. Based on the hypothetical analysis of the above scenarios, we have estimated the uncertainty that may arise from misclassification, as illustrated by the box plot C-ECP in Figure 3b. It can be observed that misclassification may lead to a more than two-fold increase in quantitative uncertainty. This indicates that the method faces challenges in quantifying multi-functional group species, and subsequent work should focus on calibrating species with more functional group types to refine classification.

Based on the above discussion, we use the interquartile range (IQR) from the uncertainty box plot in Figure 3b to evaluate the classification-based semi-quantitative method. For species with standards, the uncertainties in sensitivity are approximately 25%-50%. For species without standards, semi-quantitative sensitivity uncertainty may increase due to improper classification. This is represented by the IQR of the error classification uncertainty prediction box plot (C-ECP in Figure 3b), ranging from 25% to 80%. Additionally, our previous studies have shown that mass spectrometric sampling losses can introduce

uncertainties of approximately 10%-20% in the measurement of oxidation intermediates (Huang et al., 2019). In this study, the overall uncertainty for oxidation intermediates ranged from 30% to 85%, which is calculated as the quadrature addition of individual uncertainties.

### 3.3 Quantification of oxidation intermediates during toluene + OH reactions

The schematic diagram of the oxidation intermediates in the toluene + OH system is shown in Figure S9. Toluene rapidly

reacted with OH radicals after the light was turned on. The fates of toluene with OH radicals involved hydrogen abstraction and addition reactions (Roger and Janet, 2003). The OH hydrogen abstraction reaction was the main source of benzaldehyde. According to the oxidation mechanism of toluene (Wu et al., 2014; Schwantes et al., 2017; Wang et al., 2017; Xu et al., 2020; Vereecken, 2018; Song et al., 2021), the OH hydrogen addition reaction is the main pathway for toluene oxidation. The addition product of the toluene-OH reaction (toluene-OH adducts) has more active nonaromatic double bonds which can react with $O_2$

by hydrogen abstraction and addition reactions (Suh et al., 2002; Wu et al., 2014). Toluene-OH adducts can form an aromatic phenolic compound (cresol) by hydrogen abstraction with $O_2$, which is the dominant pathway for the formation of phenolic products (Wu et al., 2014; Ji et al., 2017). Toluene-OH adducts can react with $O_2$ over several generation cycles to form a series of bicyclic peroxy radical ($RO_2$) radicals. The bimolecular reactions of $RO_2$ radicals with NO, $RO_2$, and $HO_2$ can give rise to alkoxy radicals (RO) (Jenkin et al., 2018) and ring-retaining products ($C_7H_8O_4$, $C_7H_{10}O_4$, $C_7H_{10}O_5$, and $C_7H_8O_6$), and

then the RO radical can form ring-opening products (glyoxal, methyl glyoxal, $C_4H_4O_2$, $C_4H_4O_3$, $C_5H_6O_2$, and $C_5H_6O_3$) through a ring breakage reactions (Nishino et al., 2010; Wu et al., 2014). This study chose $C_7H_8O$, $C_7H_8O_2$, $C_7H_8O_3$, $C_4H_4O_2$, $C_4H_4O_3$, $C_5H_6O_2$, $C_5H_6O_3$, $C_7H_8O_4$, $C_7H_{10}O_4$, $C_7H_{10}O_5$, $C_7H_8O_6$ and glyoxal, methyl glyoxal as the tracers for the oxidation of toluene (Figure S9).

During the photo-oxidation process of precursors such as aromatics, I-CIMS measurements reveal that each formula may have

395 many isomers (Bi et al., 2021b). Therefore, in the semi-quantitative study of toluene oxidation products using a binding energy-based method, it is crucial to reasonably infer their structures. For the oxidation products of toluene, including $C_4H_4O_2$, $C_5H_6O_2$, $C_7H_8O_4$, $C_7H_{10}O_4$, $C_7H_{10}O_5$, and $C_7H_8O_6$,we first excluded furanones or aldehyde species that cannot be measured by I-CIMS among their isomers. Additionally, we excluded isomers originating from lower concentration multi-generation oxidation products. For example, in the toluene system, the $C_7H_8O_4I^-$ signal measured by CIMS reveals three isomers: first-generation

products in the bicyclic $RO_2$ pathway, a minor fourth-generation product hydroxyquinol derived from the phenolic pathway, and a second-generation epoxy hydroxy compound from the epoxide pathway. Laboratory experiments have revealed a negligible contribution from the epoxy pathway(Zaytsev et al., 2019), and the impact of second-generation epoxy hydroxy

compounds on the $C_7H_8O_4I^-$ signal can be considered negligible. Based on the reasonable inference above, we propose that the signals of $C_4H_4O_2$, $C_5H_6O_2$, $C_7H_8O_4$, $C_7H_{10}O_4$, $C_7H_{10}O_5$, and $C_7H_8O_6$ detected by I-CIMS primarily originate from the major first-generation products of the bicyclic $RO_2$ pathway as depicted in Figure S9. Due to their diverse functional groups, the multitype functional group species semi-quantitative equations based on the binding energy method is employed for their quantification.

For the multi-generation products $C_7H_8O_3$, $C_4H_4O_3$, and $C_5H_6O_3$, by excluding furanones and aldehydic compounds that are difficult to detect by I-CIMS, it can be inferred that the signal of $C_7H_8O_3$ primarily originates from trihydroxytoluene, the signal of $C_4H_4O_3$ primarily comes from (Z)-4-oxobut-2-enoic acid, and the signal of $C_5H_6O_3$ primarily comes from (Z)-4-oxopent-2-enoic acid and (Z)-2-methyl-4-oxobut-2-enoic acid. $C_7H_8O_3$ quantification involves semi-quantitative equations with polyphenol or diacid species, while for $C_4H_4O_3$ and $C_5H_6O_3$, which are keto acids, semi-quantitative equations incorporating multiple functional group species are used for quantification..

Furthermore, we attempted to employ voltage scanning techniques for the auxiliary identification of isomers. Isaacman et al. preliminarily explored the possible differences in the $dV_{50}$ of isomers (Isaacman-Vanwertz et al., 2018), which may serve as an important means to distinguish and quantify isomers measured by I-CIMS. In the toluene system, the $C_7H_8O$ produced during the reaction could originate from cresol in the phenolic pathway or from benzyl alcohol, a byproduct of the aldehyde pathway. Through voltage scanning, we observed a small difference in the voltage variation of $C_7H_8OI^-$ in the toluene system compared to the cresol standard samples, with $dV_{50}$ values of -0.97 and -1.12, respectively. This difference may stem from the significantly higher yield of cresol, the primary product in the toluene system, compared to benzyl alcohol (Smith et al., 1998; Baltaretu et al., 2009; Ji et al., 2017), suggesting that the influence of this type of isomerization can be disregarded during the quantification process. Therefore, $C_7H_8O$ quantification is performed using semi-quantitative equations specific to monophenol species. By comparing the voltage scanning results of $C_7H_8O_2I^-$, the oxidation products from the toluene and the dihydroxy toluene sample, and $dV_{50}$ was 0.75 and 0.72, respectively. Therefore, these results indicated that the signal for $C_7H_8O_2I^-$ could be approximated as dihydroxy toluene in the toluene system. Therefore, $C_7H_8O$ quantification is performed using semi-quantitative equations specific to polyphenol or diacid species.

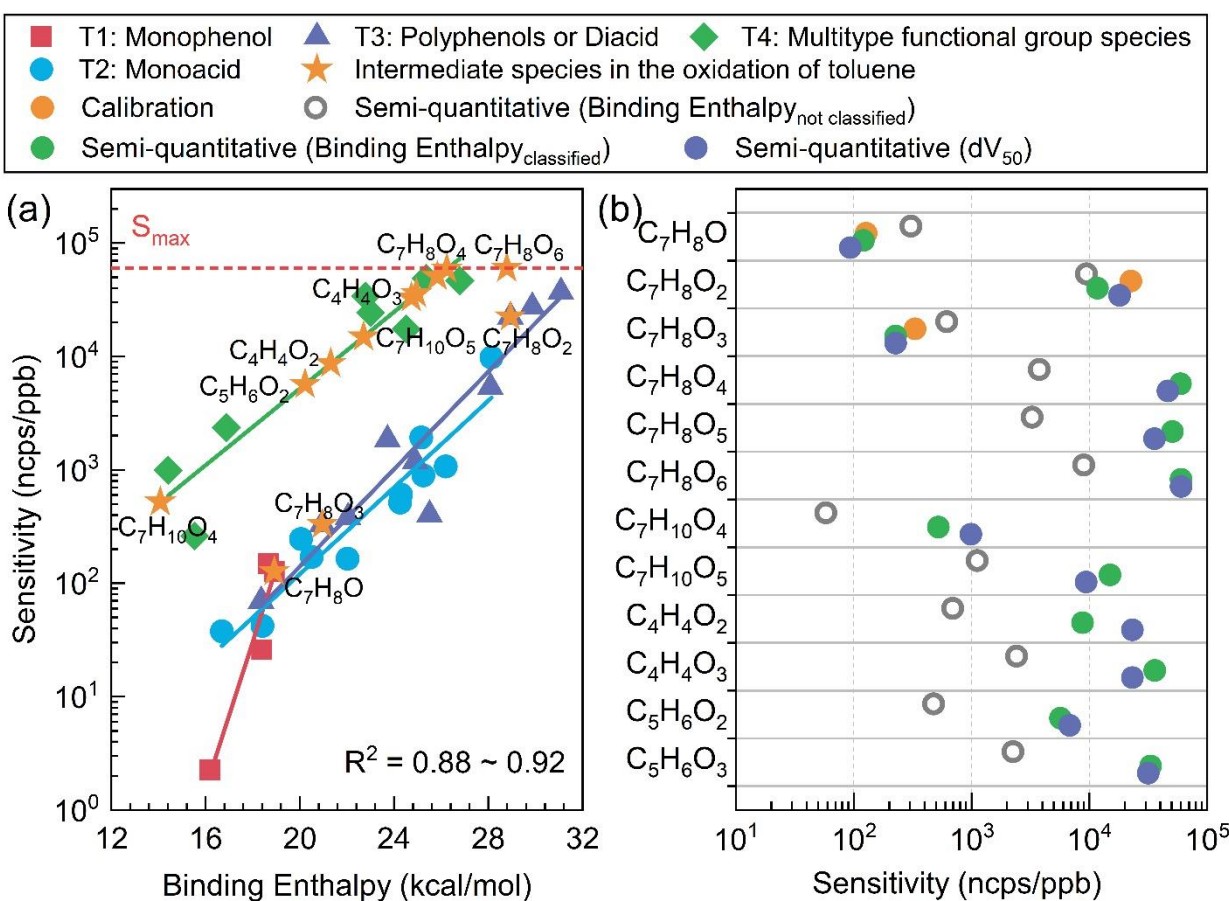

**Figure 4: (a) The sensitivity results of toluene oxidation intermediates obtained at the B3LYP/Def2TZVP (D3) level. (b) The sensitivity of key toluene oxidation intermediates obtained by direct calibration, binding energy semi-quantitative, and voltage scan semi-quantitative methods. All sensitivity values presented in the figures were acquired under the RH condition of approximately $55 \pm 5\%$.**

Based on the quantitative and semi-quantitative method established in this study, sensitivity results for key oxidation products during the toluene oxidation process were obtained, as illustrated in Figure 4. This study also utilized voltage scanning as a semi-quantitative method (Section S2 and Figure S10) to validate the classification theory calculation's semi-quantitative approach. The main difference between the two semi-quantitative methods lies in their sensitivity to the concentration of the target species and its isomers. The semi-quantitative approach based on binding enthalpy relies on the rational estimation of the structure of oxidation intermediates to obtain the sensitivity of specific products. The voltage scanning method estimates sensitivity for specific formulas but faces significant uncertainties with isomers. Furthermore, it is difficult to obtain voltage scan results for low-concentration products. This may be the reason for the difference in product sensitivity between the two semi-quantitative methods (Figure 4). These two methods are both influenced by the presence and distribution of isomers of the target species, which is also a bottleneck issue in all mass spectrometry semi-quantitative studies (Bi et al., 2021c). Hence,

it's challenging to provide an absolute assessment of the advantages and disadvantages of the two methods using current technology. As can be seen from Figure 4b, both semi-quantitative methods can be well applied to the quantification of toluene oxidation products, and they are significantly superior to the semi-quantitative method based on binding energy without classification.

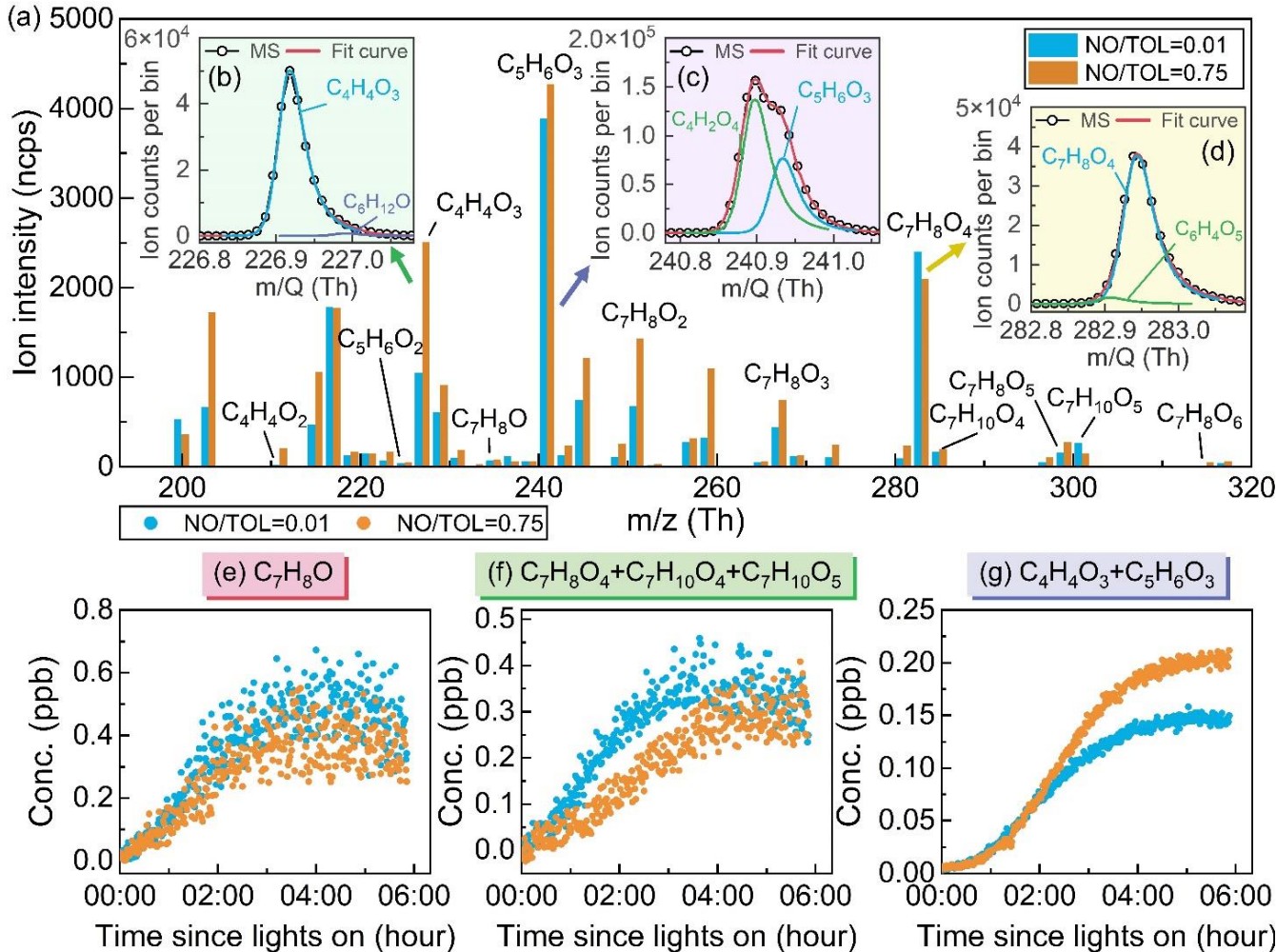

**Figure 5: (a) A one-second averaged high-resolution mass spectrum of intermediates during the oxidation of toluene. Identification of toluene oxidation products: (b) $C_4H_4O_3$, (c) $C_5H_6O_3$, and (d) $C_7H_8O_4$ in the mass spectrum. Time series of (e) $C_7H_8O$, (f) $C_7H_8O_4$ + $C_7H_{10}O_4$ + $C_7H_{10}O_5$, (g) $C_4H_4O_3$ + $C_5H_6O_3$ during the oxidation of toluene. Note: low NO condition (blue), high NO condition (orange).**

The toluene oxidation intermediates were identified through high-resolution peak fitting analysis using the TofWare software, as illustrated in Figure 5. In this study, the fitting of experimental and simulated peak shapes exhibited a high degree of correspondence ($R^2 > 0.99$), indicating that fitting species' molecular ion peak areas using high-resolution mass spectrometry can represent the measured species signal effectively. During the chamber experiment, the humidity was maintained at

455 approximately 55 ± 5%, which is consistent with the humidity condition established for the semi-quantitative equation in Figure 3, the correction factor ($RH_{corr}$) in the semi-quantitative process can be set to 1.

Time series of the oxidation products in the toluene + OH system at different concentrations of NO is shown in Figure 5 and Figure S11. After the light is turned on, the rapid generation of cresol can reach a peak of 0.4-0.5 ppb within 4 hours (Figure 5e), followed by a subsequent decline as it reacts with OH radicals. The trend of cresol formation shows little variation with
460 different NO/TOL ratios. However, there is a significant difference in the trend of formation and concentration for typical ring-retaining products, such as $C_7H_8O_4 + C_7H_{10}O_4 + C_7H_{10}O_5$, and typical ring-opening products, such as $C_4H_4O_3 + C_5H_6O_3$, glyoxal, and methylglyoxal in bicyclic peroxy radical pathway (Figure 5f-g and Figure S11). To remove the influence of precursor concentration, a ratio method was employed for comparative analysis. Under low NO conditions, the ratios of $C_7H_8O_4 + C_7H_{10}O_4 + C_7H_{10}O_5$ to $C_4H_4O_3 + C_5H_6O_3$, glyoxal, and methylglyoxal were 2.40, 0.27, and 0.16, respectively. Under NO-
465 applied conditions, the ratios of $C_7H_8O_4 + C_7H_{10}O_4 + C_7H_{10}O_5$ to $C_4H_4O_3 + C_5H_6O_3$, glyoxal, and methylglyoxal were 1.17, 0.10, and 0.09, respectively. This indicates that in the presence of high NO concentrations in the toluene oxidation system, the distribution of bicyclic $RO_2$ products tends to favour the ring-opening pathway. Additionally, the study found that the ratio of $C_5H_6O_3$ to $C_4H_4O_3$ under low NO and high NO conditions was 3.77 and 1.70, respectively. These compounds were recently identified as ring-opening products originating from the 1,5-aldehydic H-shift of alkoxy radicals during the bicyclic peroxy
radical pathway (Xu et al., 2020). The ratio of $C_5H_6O_3$ to $C_4H_4O_3$ exhibited an inverse relationship with NOx levels, similar to the case of the traditional ring-opening products $C_5H_6O_2$ to $C_4H_4O_2$ (He et al., 2023). The assumption of fixed ratios of ring-opening products in previous studies may have been due to a lack of measurement and quantification of oxidized intermediates. The impact of NO concentration on oxidation products has been a focal point in aromatic hydrocarbon oxidation studies, but this is beyond the scope of this work. It is hoped that the quantitative method proposed in this study will hold significance for
subsequent research in quantifying the influence of NO concentration on oxidation products.

## 4 Conclusion

In this study, tracers for oxidation intermediates in the toluene + OH reaction system were identified and quantified by an iodide time-of-flight chemical ionization mass spectrometer. Based on the experimental results and intermediate product characteristics of toluene oxidation system, detailed quantitative and semi-quantitative studies were conducted. Direct
quantitative methods were used to measure intermediates that had available standards, such as cresol, dihydroxy-toluene, and trihydroxy-toluene. This study directly calibrated the sensitivity of various species including monophenols, polyphenols, monoacids, diacids, phenolic acids, keto acids, furanones, etc., with sensitivity ranging from $10^0$-$10^4$ ncps/ppb, and detection limits mostly ranging from a few to 300 ppt. The study indicates a correlation between sensitivity to iodine addition and the type and quantity of polar functional groups in the species, including keto, hydroxyl, and acid groups. For species with a single
active group, sensitivity is observed to be in the order: keto group < hydroxyl group < acid group. Similarly, for species

containing more than one hydroxyl group, sensitivity increased with the addition of polar functional groups in the order: keto group < hydroxyl group < acid group.

A detailed semi-quantitative method based on binding energies was established for intermediates for which standards were not available. Given the significant selectivity differences of I-CIMS towards various reactive functional groups, this study employs a classification approach to optimize the binding energy-based semi-quantitative method. The measured species were classified into four groups: monophenols, monoacids, polyphenol or diacid species, and species with multiple functional groups for semi-quantitative analysis in the B3LYP/Def2TZVP (D3) level. The categorization method improves the accuracy of logarithmic fitting between binding energy and sensitivity, yielding $R^2$ values exceeding 0.88. This categorization method simultaneously reduces the uncertainty of semi-quantitative sensitivity. For species with standards, sensitivity uncertainties range from 25% to 50%. For species without standards, semi-quantitative sensitivity uncertainty can increase due to incorrect classification, varying from 25% to 80%. After accounting for sampling losses, the overall uncertainty for oxidation intermediates ranged from 30% to 85%. Additionally, misclassification may double quantitative uncertainty, highlighting challenges in quantifying multi-functional group species. Future research should focus on calibrating species with a greater diversity of functional groups to enhance classification accuracy.

Using the optimized I-CIMS quantitative method, quantitative measurements of intermediates formed during toluene oxidation under both low NO and high NO conditions were obtained. The research findings experimentally demonstrate the perspective that increasing NO concentration in the toluene oxidation system enhances the ring-opening pathway of bicyclic $RO_2$ products. The ratio of $C_5H_6O_3$ to $C_4H_4O_3$ was 3.77 under low NO conditions and 1.70 under high NO conditions, showing an inverse correlation with NOx levels. This contrasts with findings from traditional MCMv331 mechanism studies. Therefore, optimizing semi-quantitative methods for precise quantification of oxidation products can facilitate deeper exploration of aromatic hydrocarbon oxidation mechanisms.

This study aims to provide an optimized approach for establishing quantitative and semi-quantitative equations for measuring sensitivity with I-CIMS from a classification perspective. As this study primarily examines aromatic hydrocarbon oxidation systems, the choice of calibration species tends to prioritize aromatic hydrocarbon oxidation products or standard species with analogous functional groups. However, for other experimental systems, more experimental calibration data on species are needed for verification.

Graphical abstract

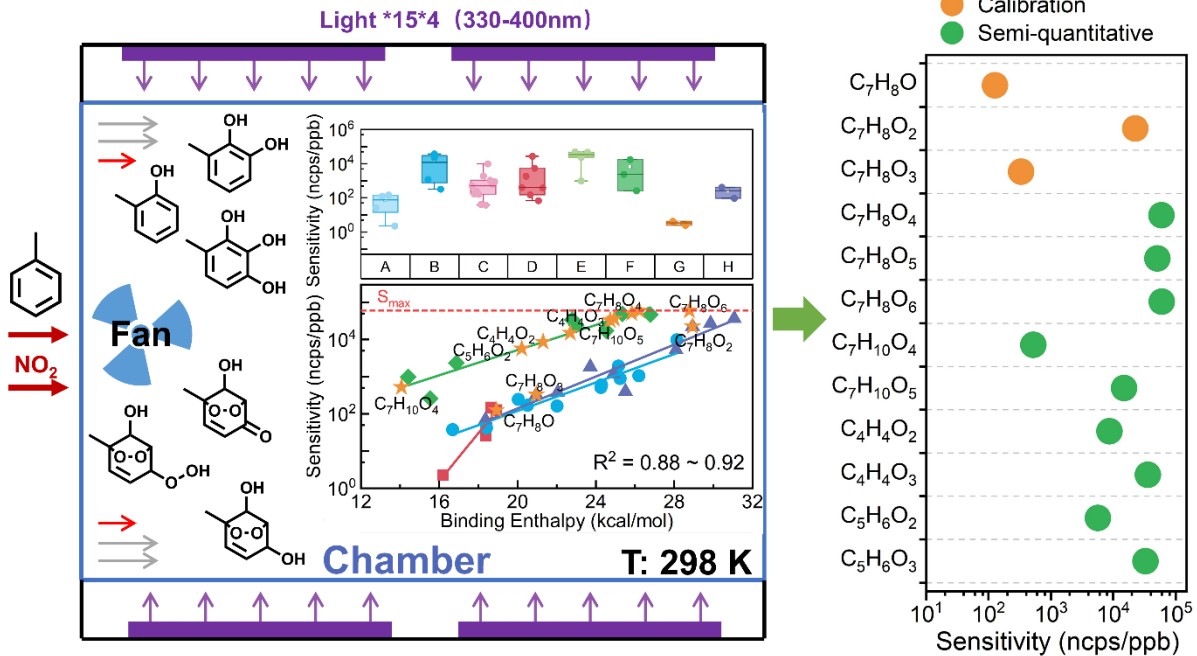

**Data availability**

The underlying research data can be accessed upon contact with the corresponding author (Xin Li: li_xin@pku.edu.cn).

**Author contribution**

X. L. and Y.L. designed and organized the experiments, supervised the data analysis, and edited the manuscript. M.S. and S.H. performed the experiments and analyzed the data. M.S. prepared the manuscript. Y.L., S.R.L., and Y.Z. contributed to the conception of this study. S.H.L., L.Z. contributed to the data analysis of VOCs measurements.

520    **Competing interests**

The contact author has declared that none of the authors has any competing interests.

**Acknowledgements**

This work was supported by the Beijing Municipal Natural Science Fund (JQ21030) and by the National Natural Science Foundation of China (Nos. 91844301, 91644108), and the National Key R&D Plan of China (grant no. 2022YFC3700201).

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
