# Peer review of "Optimizing Iodide-Adduct CIMS Quantitative Method for Toluene Oxidation Intermediates: Experimental Insights into Functional Group Differences"

_EGUsphere, 2024_

## Author Comment (AC1)

**Response to the Comments of the Reviewers**

Dear Editor and Reviewers,

We would like to thank you and the reviewers for the great efforts and elaborate work on this manuscript.

We revised the manuscript by responding to each of the suggestions in the reviews. In our response, the questions of the reviewers are shown in *Italic* form and the responses in standard form. The corresponding revisions to the manuscript are marked in blue. All updates to the original submission are tracked in the revised manuscript.

We appreciate your help and time.

Sincerely yours,

Xin Li and Co-authors.

College of Environmental Sciences and Engineering
Peking University
100871 Beijing China
E-mail: li_xin@pku.edu.cn
Tel: +86-185 1358 6831
* * *
*Manuscript Number: AMT-2024-1203.*

*Manuscript Title: Optimizing Iodide-Adduct CIMS Quantitative Method for Toluene Oxidation Intermediates: Experimental Insights into Functional Group Differences.*
* * *
**Response to Reviewer #2**

*General comments*

*In their manuscript "Optimizing Iodide-Adduct CIMS Quantitative Method for Toluene Oxidation Intermediates: Experimental Insights into Functional Group Differences," the authors present an in-depth look at understanding the sensitivity of chemical ionization mass spec using iodide as the reagent ion, and propose a method to improve calibration using calculation of binding enthalpies and classification by functional groups. Overall, I think it is a well-written paper that presents its findings well and is well grounded in the current literature. The topic is of interest to the readership of this journal and I generally support its publication here. I have one major comment/concern that needs to be addressed, but otherwise I believe it is publishable with fairly minor revisions.*

**Response:**

We would like to thank reviewer #2 for carefully reading our manuscript and for the valuable and constructive comments. We carefully revised and improved each part according to the reviewer's suggestions. Listed below are our point-by-point responses to reviewer's comments. In our response, the questions of the reviewers are shown in *Italic* form and the responses in standard form. The corresponding revisions to the manuscript are marked in blue. All updates to the original submission are tracked in the revised manuscript. Lastly, we would like to thank you for your comments and guidance.
* * *
*Major comment*

*1. The authors present a method for calibration based on binding enthalpies, an idea that has been demonstrated before but is elaborated upon nicely here. The conclusion, at times implied and at times fairly explicit, is that this approach will provide lower uncertainty that other methods, particularly the voltage scanning method currently sometimes employed. I do not dispute significant concerns and limitations of the voltage scanning method and I am not trying to make a strong pitch for it per se, but I believe the authors are somewhat too rosy about their method and have not properly discussed the limitations or provided a fair assessment of its uncertainty. Examples of my concerns in how the results are being viewed optimistically are below.*

**Response:**

We appreciate the reviewer's constructive comments. We are sorry for the unclear expression in discussing the uncertainties of the semi-quantitative method based on binding energy, which may have caused some misunderstandings among readers. The main purpose of this paper is to further optimize and categorize the semi-quantitative approach based on binding energy, rather than to imply that this method is superior to voltage scanning methods. In fact, both methods have their own limitations. In the revised manuscript, we have supplemented the discussion on the respective issues and limitations of both methods. Furthermore, we are more focused on comparing the improvement of the semi-quantitative method based on binding energy before and after classification. Now it reads as follows:

This study also utilized voltage scanning as a semi-quantitative method (Section S2 and Figure S10) to validate the classification theory calculation's semi-quantitative approach. The main difference between the two semi-quantitative methods lies in their sensitivity to the concentration of the target species and its isomers. The semi-quantitative approach based on binding enthalpy relies on the rational estimation of the structure of oxidation intermediates to obtain the sensitivity of specific products. The voltage scanning method estimates sensitivity for specific formulas but faces significant uncertainties with isomers. Furthermore, it is difficult to obtain voltage scan results for low-concentration products. This may be the reason for the difference in product sensitivity between the two semi-quantitative methods (Figure 4). These two methods are both influenced by the presence and distribution of isomers of the target species, which is also a bottleneck issue in all mass spectrometry semi-quantitative studies (Bi et al., 2021b). Hence, it's challenging to provide an absolute assessment of the advantages and disadvantages of the two methods using current technology. As can be seen from Figure 4, both semi-quantitative methods can be well applied to the quantification of toluene oxidation products, and they are significantly superior to the semi-quantitative method based on binding energy without classification.

Furthermore, based on the reviewer's suggestions, we have thoroughly discussed the limitations of the semi-quantitative method based on the classified binding energy and provided an in-depth evaluation of its uncertainties (refer to the response to sub-question (1.2) in the reviewer's major comment). Below are our detailed responses to each sub-question in the reviewer's major comment.
* * *
*1.1 Line 301. In Figure S6, there are a number of compounds with low binding energies and high sensitivities. These do not seem to be any of the classified compounds (in Figures 6b-6e, none go down to enthalpies below 10). These points seem to be driving a lot of the poor correlation, so is the observed improvement in R2 because of a true improvement, or more becomes some outliers seem to be excluded. It is also a little hard to tell, but from Figure S6 it looks like most things are on the same line, except the multi-functional compounds, is this correct? In other words, if you just exclude the multi-functional compounds from Figure 6a and do the fit, do you get a similarly high R2? I'm not sure that would negate some of the conclusions of the authors, but it seems an interesting fact if I am interpreting correctly.*

**Response:**
We appreciate the reviewer's comments. In this study, we employed a correlation fitting approach that categorized species excluding furanones and those with binding energies lower than 10. The exclusion of furfural and 3-methyl-2(5H)-furanone is due to their low sensitivity and high detection limits (Table S1), indicating that I-CIMS does not have an advantage in measuring furanones. The reason for excluding species with binding energies lower than 10 is that I-CIMS exhibits a minimum sensitivity threshold. When the binding energy between the analyte species and I- approaches or falls below the binding energy of $H_2OI^-$ at 12 kcal/mol at the B3LYP/Def2TZVP (D3) level, the instrument cannot detect the species.

Furthermore, when we excluded furfural and species with binding energies lower than 10 from the fitting in Figure S6a, we found that the R2 values remained poor, with values of 0.34, 0.46, and 0.52 respectively at the PBE/SDD, PBE/SDD (D3), and B3LYP/Def2TZVP levels (Figure S6). This indicates that the improvement in R2 is attributed to the categorization rather than the exclusion of outliers.

To clearly demonstrate the improvement achieved through the categorical fitting approach, we have maintained consistency in the fitted species before and after categorization in the revised Figure S6. The revised Figure S6 is presented as follows:

[Figure]

**Figure S6: Fitting curve for cluster binding enthalpies and logarithmic sensitivities at PBE/SDD, PBE/SDD (D3), and B3LYP/Def2TZVP level**

Additionally, excluding multifunctional compounds from the fitting in Figure 6a indeed improved the $R^2$ values, with respective $R^2$ values of 0.49, 0.65, and 0.89 at the PBE/SDD, PBE/SDD (D3), and B3LYP/Def2TZVP levels. However, when other evaluation parameters are taken into account (Table R1), the effect of classification becomes more pronounced (Figure R1). However, considering that the classification of monophenols, monoacids, and polyphenols, as well as diacids, is relatively straightforward and clear, and there are differences in the sensitivity of these species to humidity response. Therefore, we recommend a more detailed classification approach, which is likely to be in greater demand in future research. With the advancements in subsequent calibration techniques, it may be possible to explore the changes in sensitivity among various types of functional group species.

**Table R1 Evaluation parameters and calculation methods for linear regression**

| Abbreviation | English Name | Calculation Method |
|---|---|---|
| FB | Fractional bias | $100\% * \dfrac{2}{N} \sum \dfrac{(Mod_i - Exp_i)}{(Mod_i + Exp_i)}$ |
| FE | Fractional error | $100\% * \dfrac{2}{N} \sum \dfrac{|Mod_i - Exp_i|}{(Mod_i + Exp_i)}$ |
| NMB | Normalized mean bias | $100\% * \dfrac{\sum(Mod_i - Exp_i)}{\sum(Exp_i)}$ |
| NME | Normalized mean error | $100\% * \dfrac{\sum|Mod_i - Exp_i|}{\sum(Exp_i)}$ |
| $R^2$ | Coefficient of determination | $1 - \left( \dfrac{\sum(Exp_i - Mod_i)^2}{\sum(Exp_i - \overline{Exp})^2} \right)$ |

[Figure]

**Figure R1 The multi-parameter evaluation of the linear regression**
* * *
*1.2 Line 308-314. This comparison in uncertainty is unfair, in part because it is fairly circular and in part because it is optimistic. Essentially, in this study, a calibration curve is made between binding enthalpy and sensitivity for subsets of compounds, then that calibration is reassigned to the same compounds and found to not have a lot of error. It is true this would not work well if the correlation between binding enthalpy and sensitivity were poor, but that is all this shows, which the R2 has already shown (consider for example, mono-phenols, for which a line is drawn for only 4 points, then this line is reapplied to these four points to demonstrate low uncertainty). This also depends on selecting the correct classification for the mass, as it is demonstrated in Figure 3 and S6 that without classification there is a wide range. It's not clear to me how to assign a classification to compounds that were not introduced by standards (nor is it really discussed in detail). A third paper by Bi et al. in 2021 that I think is not cited here (10.5194/amt-14-3895-2021) suggests that photolysis of most precursors, including aromatics, produces many isomers for each formula. How would one go about assigning the proper classification to an ion in the absence of standards, when it is possible that the ion contains multiple distinct structures which may have different functional groups? Of course, all methods have their limitations, but the 40% estimate presented here is a best case that really only applies for compounds that were used to generate the calibration curve. I don't understand how the 40% could be considered an upper limit (it seems to me an upper limit would be caused by mis-assigning classification, in which case Figure 3 implies the error could be as high as 2 orders of magnitude).*

**Response:**

We appreciate the reviewer's constructive comments, and we fully acknowledge that the mis-assigning classification can significantly introduce uncertainties into this method. In the revised manuscript, we have added a discussion on the classification of oxidized products without standard samples and conducted a detailed analysis of the potential impacts of mis-assigning classification. Now it reads as follows:

The semi-quantitative uncertainty was computed by dividing the absolute difference between the measured sensitivity and the calculated sensitivity by the measured sensitivity, as illustrated in Figure 3. For standard samples, the findings indicated a satisfactory concordance between the calculated and

experimental sensitivity factors, with relative deviations below 40% (Figure 3a). As can be observed from Figure 3b, the classified semi-quantitative method based on the binding energies (C-SS in Figure 3b) enhances the accuracy of quantification.

However, for species without standard samples, the selection of classification becomes paramount, as inappropriate classification may introduce significant uncertainties. In this research, since the classification features of monophenols and monoacids are quite distinct, we can disregard the errors caused by misclassification. For the third and fourth groups of species, a more detailed classification is possible, distinguishing them into polyphenol, diacid, phenolic acid, and keto acid species. This refined classification leads to new semi-quantitative relationships, as illustrated in Figure S8. Taking the fourth category of multitype functional group species as an example, when we assume that it contains only phenolic acid species and erroneously apply this equation to quantify keto acid species, it will introduce significant uncertainties due to misclassification. Furthermore, when the multitype functional group species contain two acid groups and one hydroxyl group, the uncertainty arising from their misclassification as diacid species also needs to be taken into consideration. Based on the hypothetical analysis of the above scenarios, we have estimated the uncertainty that may arise from misclassification, as illustrated by the box plot C-ECP in Figure 3b. It can be observed that misclassification may lead to a more than two-fold increase in quantitative uncertainty. This indicates that the method faces challenges in quantifying multi-functional group species, and subsequent work should focus on calibrating species with more functional group types to refine classification.

[Figure]

**Figure 3: (a) The difference between the measured sensitivity and the calculated sensitivity for standards at the B3LYP/Def2TZVP (D3) level. (b) The uncertainty of classification-based semi-quantitative methods at the B3LYP/Def2TZVP (D3) level. The uncertainty is computed by dividing the absolute difference between the measured sensitivity and the calculated sensitivity by the measured sensitivity. All sensitivity values presented in the figures were acquired under the RH condition of approximately 55 ± 5%.**

[Figure]

**Figure S8: Fitting curve for cluster binding enthalpies and logarithmic sensitivities of polyphenol, diacid, keto acid, and phenolic acid species at the B3LYP/Def2TZVP (D3) level. All sensitivity values presented in the figures were acquired under the RH condition of approximately 55 ± 5%.**

Based on the above discussion, we use the interquartile range (IQR) from the uncertainty box plot in Figure 3b to evaluate the classification-based semi-quantitative method. For species with standards, the uncertainties in sensitivity are approximately 25%-50%. For species without standards, semi-quantitative sensitivity uncertainty may increase due to improper classification. This is represented by the IQR of the error classification uncertainty prediction box plot (C-ECP in Figure 3b), ranging from 25% to 80%. Additionally, our previous studies have shown that mass spectrometric sampling losses can introduce uncertainties of approximately 10%-20% in the measurement of oxidation intermediates (Huang et al., 2019). In this study, the overall uncertainty for oxidation intermediates ranged from 30% to 85%, which is calculated as the quadrature addition of individual uncertainties.

Regarding the classification of intermediate oxidation products and isomers, we have added a detailed explanation in the application section of the quantitative method (Section 3.3). Now it reads as follows:

[revised manuscript text omitted]

**Response:**

We appreciate the reviewer's comments. The study found that when we grouped diacid or polyphenol species with multiple functional group species, the fit of the linear regression significantly deteriorated, with an $R^2$ value of only 0.41. This indicates significant differences between these two categories of species. Further analysis, using citric acid as an example, shows it has three carboxyl groups and one hydroxyl group. In this study, we classified it as a species with multiple different functional groups, resulting in an $R^2$ of 0.88 and an uncertainty of 41%. However, when we reclassified it as a diacid or polyphenol species, the $R^2$ weakened to 0.62, and the uncertainty increased to 88%. This suggests that species containing even one different type of functional group are not suitable for classification as diacid or polyphenol species. We hypothesize that dicarboxylic acids or polyphenols, due to their stereochemical features and spatial hindrance effects, may exhibit interactions between hydrogen bonds of the same type. In contrast, multifunctional compounds can form different types of hydrogen bonds with I, thereby demonstrating their sensitivity to a given binding enthalpy. Therefore, we preliminarily define multifunctional functional groups as those that refer specifically to a variety of different functional groups. We propose that the diacid or polyphenol species category should consist of species with all the same functional groups. If species with different functional groups are present, they should be preferentially classified into a multiple functional group species category.

In the revised manuscript, we have added the following content to clarify the applicability of the classifications. Now it reads as follows:

Line 341-351 in tracked changes manuscript: For species with multiple functional groups, primarily consisting of phenolic acids and keto acids, examples include salicylic acid, citric acid, and lactic acid, where iodide-adducts tend to form two different hydrogen bonds (Figure 2d). The fitting performance ($R^2$) for the correlation between binding enthalpies and sensitivity is 0.88 (Figure 3d). The sensitivity of these compounds is significantly higher than that of other categories, with binding enthalpies to iodide ion ranging between 15 to 27 kcal/mol. Among them, citric acid is quite unique, as it has three carboxyl groups and one hydroxyl group. When reclassified as a diacid or polyphenol species, the $R^2$ weakened to 0.62, and the relative deviations between measured sensitivity and calculated sensitivity increased more than twofold, reaching 88%. This indicates that species containing even one different type of functional group should preferably be classified into the multiple functional groups category.

Based on the results from existing standards, categorizing into monophenols, monoacids, polyphenol or diacid species, and species with multiple functional groups allows for improved semi-quantitative analysis. Perhaps future research could involve synthesizing or customizing species with a greater variety of functional groups for quantitative studies. This would help explore the sensitivity of species with combinations such as two acid groups and one carbonyl group, or multiple carbonyl groups and one acid group, to achieve more refined classification.
* * *
*1.4. Line 349. What do the authors mean less differences? Are they again referring to the comparison to calibrants? As an example of my concern, see Figure S10. How is relative sensitivity assigned here? For example, there are 3 points that fall noticeably off the line at dV50= 4, 5.5, and 6.5. One explanation is that dV50 does a bad job of capturing sensitivity. The other explanation that does not seem to be considered here, is that dV50 is correctly estimating sensitivity, and the relative sensitivity estimated by the binding enthalpy approach is incorrect (e.g., assigned to the wrong classification). To be very clear, I do not mean to imply either case is correct, but rather that there are limitations to the proposed method developed here that are not really being considered, and could be providing spurious understanding. 4.*

**Response:**

We apologize for the unclear labeling of the figures. In Figure S10, the relative sensitivity indicated refers to sensitivity relative to the maximum sensitivity, not compared to the binding energy-based semi-quantitative method. Therefore, the three points at $dV_{50} = 4$, 5.5, and 6.5 are only used to fit data points for the voltage scanning-based semi-quantitative method. We have revised Figure S10 in the revised manuscript and added Figure S10a to demonstrate the calculation method for relative sensitivity in $dV_{50}$. The revised Figure S10 is depicted as follows:

[Figure]

**Figure S10: (a) Fitting curves of species sensitivity relative variation under scanning voltage; (b) Fitting results of the relative binding energy indicator dV50 for the iodide adducts of standard species and aromatic hydrocarbon oxidation products with the species sensitivity relative to maximum sensitivity, where dV50 represents the voltage at half signal maximum.**

Furthermore, we are more focused on comparing the improvement of the semi-quantitative method based

on binding energy before and after classification. Therefore, we have carefully revised this statement in the revised manuscript. Now it reads as follows:

This study also utilized voltage scanning as a semi-quantitative method (Section S2 and Figure S10) to validate the classification theory calculation's semi-quantitative approach. The main difference between the two semi-quantitative methods lies in their sensitivity to the concentration of the target species and its isomers. The semi-quantitative approach based on binding enthalpy relies on the rational estimation of the structure of oxidation intermediates to obtain the sensitivity of specific products. The voltage scanning method estimates sensitivity for specific formulas but faces significant uncertainties with isomers. Furthermore, it is difficult to obtain voltage scan results for low-concentration products. This may be the reason for the difference in product sensitivity between the two semi-quantitative methods (Figure 4). These two methods are both influenced by the presence and distribution of isomers of the target species, which is also a bottleneck issue in all mass spectrometry semi-quantitative studies (Bi et al., 2021b). Hence, it's challenging to provide an absolute assessment of the advantages and disadvantages of the two methods using current technology. As can be seen from Figure 4, both semi-quantitative methods can be well applied to the quantification of toluene oxidation products, and they are significantly superior to the semi-quantitative method based on binding energy without classification.
* * *
**Technical comments**

*1. Line 38. "identify" isn't really the right word here. It has been shown (e.g., Riva et al. 2019, Isaacman-VanWertz et al. 2018, as cited) that current CIMS can see essentially all the reactive organic compounds, but they are generally only classified by molecular formula, not really identified, which to me implies some knowledge of molecular structure. This applies at line 100 as well, where I think "identified" should be changed to "classified by exact molecular weight.*

**Response:**
Revised accordingly. Now it reads as follows:
*Line 38:* Chemical ionization mass spectrometry (CIMS) techniques allow classification based on exact molecular weight for nearly all semi-volatility and low volatility intermediate species.
*Line 100:* Then, the mixed flow was passed through an orifice into the high-resolution time-of-flight mass spectrometry and arrived at the detector to be classified by exact molecular weight.
* * *
*2. Line 59. Typo on semi-quantitative.*

**Response:**
Revised accordingly.
* * *
*3. Line 60. "Discovered" should not be capitalized.*

**Response:**
Revised accordingly.
* * *
*4. Line 103: "and making the absolute" is incorrect grammar and should be corrected.*

**Response:**

We are sorry for the mistake. We replaced "making" with "ensured that". Now it reads as follows:

For mass spectrometry analysis, we used the single-ion peaks for $I^-$, $H_2OI^-$, $HNO_3I^-$, and $I_3^-$ for mass calibration, and ensured that the absolute in-flight deviation of the m/Q was below 5 ppm (2σ), which was much lower than the instrument guideline of 20 ppm.
* * *
*5. Section 2.2. The order of this section is a little confusing. In the beginning of the first paragraph is a discussion of what is "often" done for calibration, then a mention of humidity, and only then is it made clear what is being done here. Reorganize so that a description of the actual calibration system comes right after the description of what is often done.*

**Response:**

We appreciate the reviewer's comments, and we have carefully reorganized this part in the revised manuscript. Now it reads as follows:

In this study, 37 species with different functional group were directly calibrated using certified penetrant tubes (CPT) and a home-built liquid calibration unit (LCU), including monophenols, monoacids, polyphenols, diacids, phenolic acids, keto acids, furanones, and other species (Table S1).

For OVOCs that can be customized to a standard gas and penetrant tubes, calibration is often performed using certified penetrant tubes (KinTek Inc.) at 5-6 gradient concentration levels (Huang et al., 2019). The calibrated concentration ranges from dozens of ppt to several ppb levels. However, because many standard samples are liquid or solid, it is challenging to make permeable tubes that have stable permeability. Because this study focused on the gas phase reaction, an appropriate home-built liquid calibration unit was designed so the OVOCs could be calibrated in the gasous form under normal temperature and pressure. The standard sample was mixed with a soluble solvent, including water, dichloromethane, or acetone, and the solvent was atomized at a given flow rate. Subsequently, the atomized gas was mixed with high-flow nitrogen to ensure the complete evaporation of the atomized droplets, which were then injected into the sampling port. No liquid condensation was observed on the wall of the mixing unit, and no particulate matter was present. After sufficient equilibration time, stable signals of standard samples could be detected in I-CIMS. The specific calibration method can be found in a study by Qiu et al. (Qiu et al., 2021) and Qu et al. (Qu et al., 2023).

To investigate the influence of humidity on calibration, both CPT and LCU calibration system are equipped with a humidification section that can control humidity within the range of 0.12 to 22.00 mmol/mol, which corresponds to a relative humidity (RH) of 0.4% to 70% at a temperature of 25°C and a pressure of 101.325 kPa. During the calibration process, by adjusting the humidity, the changing relationship between the sensitivities of various standard samples and the water vapor pressure can be obtained.

In this study, the molecular weight range of directly calibrated species was 46.01 to 216.17, which covered the molecular weight range of the principal gaseous intermediates of toluene (48.04-203.15). The linear correlation between the normalized signal values of the directly calibrated species and the concentration was excellent, with $R^2$ values greater than 0.99 for most species. For species whose sensitivities could be directly calibrated, concentrations can be calculated using Equation (2):

$$[X\_ppb] = \frac{\text{Normalized signal}}{\text{Sensitivity from direct quantification} \times RH_{Corr}} \quad (2)$$

where $RH_{Corr}$ represents the humidity correction equation. The humidity correction equations for various

functional group standard samples are provided in Section S1 and illustrated in Figure S2 of the Supplement. Additionally, the humidity correction for species without standard samples can be estimated based on the characteristics of species with similar functional groups.

Assuming that the random uncertainty of the CIMS detector counts follows Poisson statistics, the signal-to-noise ratio (S/N) and the detection limits for calibrated species can be calculated using Equation (3) (Bertram et al., 2011):

$$\frac{S}{N} = \frac{C_f[X]t}{\sqrt{C_f[X]t + 2Bt}} \tag{3}$$

Where [X] represents the detection limits (ppbv), $C_f$ is the sensitivity factor from calibration (ncps s$^{-1}$ ppbv$^{-1}$), t is the integration time (s), B represents the background normalized signal rate (ncps s$^{-1}$). In this study, we calculate the detection limits for all 37 calibrated species under 1 second averaging and a signal-to-noise ratio of 3. The data results are listed in Table S1.
* * *
*6. Line 126. What is the humidity correction equation? Similarly, in line 145, what is the mass transmission correction equation? Are these listed in the SI somewhere that I missed? I gather it is like the data in Figure S4, but some mention should be made here.*

**Response:**

We appreciate the reviewer's comments, we have added more information regarding the humidity correction equation and transmission correction equation immediately following Equations (2) and (4), respectively. Additionally, we have explained the use of humidity correction equation and transmission correction equation in the " Result and discussion " section. Now it reads as follows:

Line 146-152 in tracked changes manuscript: For species whose sensitivities could be directly calibrated, concentrations can be calculated using Equation (2):

$$[X\_ppb] = \frac{Normalized\ signal}{Sensitivity\ from\ direct\ quantification \times RH_{Corr}} \tag{2}$$

where $RH_{Corr}$ represents the humidity correction equation. The humidity correction equation, detailing the changing relationship between sensitivities of various standard samples and water vapor pressure, is presented in Section S1 and visually illustrated in Figure S2 of the Supplement. Additionally, the humidity correction for species without standard samples can be estimated based on the characteristics of species with similar functional groups.

Line 272-281 in tracked changes manuscript: Humidity has a significant influence on the sensitivity of iodine adducts (Ye et al., 2021; Lee et al., 2014). Through the establishment of humidity-dependent parametric equations, species sensitivity under different humidity conditions can be obtained. As elaborated in Section S1, this study established humidity-dependent parametric equations for four categories of compounds (Figure S2): (1) single active functional group compounds like acrylic acid, which show rapid sensitivity decline with increasing humidity, (2) multiple active functional group compounds like pinonic acid, which have higher sensitivity and are less affected by humidity, (3) polyphenol compounds like 2,4,6-trihydroxytoluene, which are nearly unaffected by humidity, and (4) small-molecular-weight acids like formic acid, which show increased sensitivity at low humidity but decreased sensitivity at higher humidity. These humidity-dependent parametric equations correspond to $RH_{Corr}$ in Equation (2), (4), and (S2).

Line 168-173 in tracked changes manuscript: These four factors are combined to generate a detailed semi-quantitative expression, as shown in Equation (4):

$$[\text{X\_ppb}] = \frac{\text{Normalized signal}}{\text{Sensitivity from binding enthalpy} \times \text{MassTrans} \times \text{RH}_{\text{Corr}}} \quad (4)$$

where MassTrans represents the mass transmission correction equation. The mass transmission correction equation characterizes the ability of the mass spectrometer to introduce ions with different mass-to-charge ratios from the IMR to the mass detector (Heinritzi et al., 2016). A detailed information can be found in Section 3.2.

Line 288-295 in tracked changes manuscript: For the toluene oxidation system under investigation, this study analyzed the mass transmission effects of species within the mass range (180-350 m/z), where the primary gaseous oxidation products of toluene are located (Figure S5). It is shown that within the specified mass-to-charge ratio range, the mass discrimination effect has minimal influence on the sensitivity of the target species. This impact remains similarly negligible when comparing with mass transmission curves from prior studies (Ye et al., 2021) in the 180-350 m/z range. Therefore, when quantifying toluene oxidation products, the correction factor (MassTrans) of Equation (4) in the semi-quantitative process can be set to 1. However, for species with higher mass-to-charge ratios, it is crucial to account for mass correction using the mass transmission curves reported by Heinritzi et.al (Heinritzi et al., 2016) and Ye et. al (Ye et al., 2021).
* * *
*7. Line 158. Typos in this phrase.*

**Response:**

We are sorry for the typo. It should be "these" rather than "there". Now it reads as follows:

Therefore, all geometrical optimization of standard species, products, and iodide ions was performed using these theoretical methods: PBE/SDD (Schaefer, 2013), PBE/SDD-D3, and B3LYP/Def2TZVP-D3 (Weigend and Ahlrichs, 2005; Weigend, 2006) levels.
* * *
*8. Line 176. How did the wall of the chamber form HONO in the NO-free experiments?*

**Response:**

We are sorry for the confusing expression. In this paper, the NO-free condition refers to the experimental process in the smog chamber without artificial addition of NO. However, despite the rigorous purification process, trace amounts of NOx may still persist in chamber. In this study, following the comprehensive purification process, the residual concentrations of NOx within the chamber remained notably low, specifically measuring below 0.4 ppb, which approaches the detection thresholds of the commercial chemiluminescence instrument (Thermo Scientific™ Model 42i). These values are in close alignment with the findings reported in reviews of analogous chambers conducted by Chu et al. (Chu et al., 2022). Due to the inevitable presence of background NOx concentrations, the heterogeneous reaction of $NO_2$ on the Teflon surface within the chamber results in the release of nitrous acid (HONO) (Chu et al., 2022; Rohrer et al., 2005; Li et al., 2019). In this study, the chamber's HONO background concentration was approximately 0.2 ppb under wet conditions (RH ≈ 60%). Subsequently, upon illumination, the OH radical generated through the photolysis of HONO serves as a crucial source of free radicals. The photolysis of HONO led to the formation of OH radicals ranging from $1.23 \times 10^6$ molecule cm$^{-3}$ to $3.55 \times 10^6$ molecule cm$^{-3}$, which triggered the atmospheric oxidation reaction of aromatic hydrocarbons.

In the revised article, we have altered the inappropriate expression of "NO-free." The revised manuscript now utilizes "low NO and high NO condition" to replace "NO-free and NO-applied conditions." Additionally, in section 2.4, we have provided a detailed description of the HONO sources within the chamber and clarified the NO and toluene concentration scenarios corresponding to the low and high NO conditions. Now it reads as follows:

Before the experiment, the chamber was cleaned with 100 L/min dry synthetic air (made from liquid $N_2$ and $O_2$ with a ratio of 80:20, purity > 99.999%) for at least 10 hours. The relative humidity of the chamber was humidified to approximately $55 \pm 5\%$, while the temperature was maintained at $26 \pm 1$ °C. Following the comprehensive purification process, the residual concentrations of NOx within the chamber were notably low, specifically measuring under 0.4 ppb, nearing the detection thresholds of the commercial chemiluminescence technology instrument (Thermo Scientific™ Model 42i). Due to the inevitable presence of background NOx concentrations, the heterogeneous reaction of $NO_2$ on the Teflon surface within the chamber results in the release of nitrous acid (HONO) (Chu et al., 2022; Rohrer et al., 2005; Li et al., 2019). Then, VOC precursor and NO were introduced into the chamber, leading to initial toluene concentrations of approximately 100 ppbv (without NO injection) and 80 ppbv (with 60 ppb NO) in the chamber. The initial NO/toluene ratios were 0.01 and 0.75, respectively, corresponding to the low NO and high NO conditions in this study. After the chamber air became stable (within 10-20 min), the lights were turned on. Subsequently, upon illumination, the OH radical generated through the photolysis of HONO serves as a crucial source of free radicals. The photolysis of HONO led to the formation of OH radicals ranging from $1.23 \times 10^6$ molecule cm$^{-3}$ to $3.55 \times 10^6$ molecule cm$^{-3}$, which triggered the atmospheric oxidation reaction of aromatic hydrocarbons.
* * *
*9. Figure 1. Bar charts (and stacked charts) cannot be used with logarithmic axes because there is no real zero so the bottom of the axis is arbitarily selected. The visual size of each bar does not actually represent relative difference. For example, one could set the axis to start at 10^-10 and then all the bars look basically the same. Similarly, bars 2 and 3 differ by 100 units, while pars 5 and 6 differ by 10,000 units, yet the difference in their relative areas is the same. This figure should be remade as a scatter plot.*

**Response:**
We appreciate the reviewer's comments, and we have remade Figure 1 as a scatter plot in the revised manuscript. Now it shows as follows:

[Figure]

Figure 1: (a) Direct quantitation sensitivity results of 37 standard materials (b) Sensitivity statistics for standard materials containing different functional groups (c)-(f) Calibration curves of 2, 4-dihydroxytoluene ($C_7H_8O_2$), formic acid ($CH_2O_2$), salicylic acid ($C_7H_6O_3$), and levulinic acid ($C_5H_8O_3$). Note. Details of species with corresponding serial numbers in figure (a) are available in TableS1.
* * *
*10. Line 191-192. Which of these bars or points are these three named compounds? The reference to the figure implies I should be able to tell, but they are not labeled, and two of the three of them are not mentioned in the caption.*

**Response:**

We are sorry for the confusing expression. The names of the compounds can be found in Table S1 according to the sequence number in Figure 1a. However, in order to make the manuscript more readable, we have added chemical formulas after the species names, making it possible to quickly find them in Figure 1a. Now it reads as follows:

For toluene oxidation intermediates with standards, such as m-cresol ($C_7H_8O$), 2,4-dihydroxytoluene ($C_7H_8O_2$), and 2,4,6-trihydroxytoluene ($C_7H_8O_3$), the directly calibrated sensitivity in I-CIMS is $1.3 \times 10^2$ ncps/ppb, $2.2 \times 10^4$ ncps/ppb, and $3.3 \times 10^2$ ncps/ppb, respectively (错误!未找到引用源。 and Table S1).
* * *
*11. Line 197. I'm a bit confused. Are the compounds shown in Figure 1 those that are lacking standards? If so, how are these categorized by structure? This sentence, and indeed this paragraph, somewhat confuses me about which points are those lacking standards and which are for standards, and how the*

*former is being classified. Edit this paragraph for clarity around known and unknown compounds.*

**Response:**

We are sorry for the confusing expression. All the compounds shown in Figure 1 are standard samples, which are selected based on the characteristics of toluene oxidation products. The purpose is to make the establishment of the semi-quantitative equation more suitable for the subsequent study of intermediate products in toluene oxidation system.

We have carefully revised this paragraph in the revised manuscript. Now it reads as follows:

For toluene oxidation intermediates with standards, such as m-cresol ($C_7H_8O$), 2,4-dihydroxytoluene ($C_7H_8O_2$), and 2,4,6-trihydroxytoluene ($C_7H_8O_3$), the directly calibrated sensitivity in I-CIMS is 1.3 × $10^2$ ncps/ppb, 2.2 × $10^4$ ncps/ppb, and 3.3 × $10^2$ ncps/ppb, respectively (Figure 1 and Table S1). In the case of toluene oxidation intermediates lacking standards, this study selects standard samples with similar reactive functional groups to toluene oxidation intermediates for calibration, so that the subsequent quantitative and semi-quantitative equations are more applicable to the toluene oxidation system. Typical oxidation products of toluene include aromatic phenols, ring-retaining phenols, ring-opening acids, ring-opening keto acids, ring-opening phenolic acids, and ring-opening furanones, among others (He et al., 2023). Based on the characteristics of the aforementioned toluene oxidation intermediates, this study selected 37 standard samples for calibration, including the main types of monophenols, polyphenols, monoacids, diacids, phenolic acids, keto acids, and furanones, as shown in Figure 1 and Table S1.
* * *
*12. Line 206-207. The described increase in sensitivity is not really clear to me from Figure 1, as reference. Which compounds contain only a single active keto group? There is no indication of that in the figure. I guess maybe they mean the furanones? Although, really one of those is an ester group, not a ketone, and the structure of furanones is never described so many readers may not realize these are ketones.*

**Response:**

We are sorry for the confusing expression. At the beginning of this statement, we indicated that furanones were used to represent the species type that only contained active ketone groups. Now it reads as follows:

As depicted in Figure 1b, the sensitivity of furanones, monophenols, and monoacids gradually increases, indicating that species with a single active group, such as keto, hydroxyl, and acid groups, exhibit increasing sensitivity in the order listed.
* * *
*13. Line 208-216. The number of significant digits on sensitivities seems optimistic, is it really known to 6 digits for e.g., allylacetic acid? I think limiting it to maybe 2 would be more realistic, given the amount of uncertainty in these instruments.*

**Response:**

We appreciate the reviewer's comments, and we have standardized the significant digits of the sensitivity in the manuscript and Table S1 to 2 digits according to the reviewer's suggestions. Now it reads as follows:

Among them, furanone containing keto groups were the least sensitive, and the sensitivities of furfural and 3-methyl-2(5H)-furanone were 3 ncps/ppb and 4 ncps/ppb, respectively. The detection limits for furfural and 3-methyl-2(5H)-furanone are also very high (Table S1), indicating that I-CIMS does not have an advantage in measuring furanones. The sensitivities of monophenolic compounds such as phenol

and m-cresol were $1.5 \times 10^2$ ncps/ppb and $1.3 \times 10^2$ ncps/ppb, respectively. I-CIMS demonstrates good detection capability for phenol and m-cresol, with low detection limits of 0.11 and 0.08 ppb (in 1-second, S/N=3), respectively. However, it exhibits relatively lower sensitivity for larger mass compounds such as 2,6-xylenol and texanol, resulting in higher detection limits. Previous studies have also shown that I-CIMS has good sensitivity toward compounds containing carboxylic acid groups (Mcneill et al., 2007; Le Breton et al., 2012; Lee et al., 2014). Similarly, here we found that the sensitivity of monoacids was higher, and the sensitivities of formic acid, allylacetic acid, and 2-ethylhexanoic acid were $1.9 \times 10^3$ ncps/ppb, $1.1 \times 10^3$ ncps/ppb, and $8.9 \times 10^2$ ncps/ppb, respectively. I-CIMS exhibits low detection limits for monoacids, ranging from a few to 400 ppt (Table S1), enabling the detection of species at the molecular level.
* * *
*14. Line 218. What do they authors mean by "detection of species at the molecular level"?*

**Response:**

We are sorry for the inaccurate expression. We have removed the redundant and incorrect statement in the revised manuscript. Now it reads as follows:

I-CIMS exhibits low detection limits for monoacids, ranging from a few to 400 ppt (Table S1).
* * *
*15. Line 255. "taking the phenolic pathway for toluene" is incorrect grammar.*

**Response:**

We are sorry for the inaccurate expression. We have carefully revised this statement in the revised manuscript. Now it reads as follows:

For example, when examining the phenolic pathway for toluene, the yield of cresol obtained through semi-quantitative analysis was 2.5 times lower than that obtained through direct calibration.
* * *
*16. Line 260. For compounds with no known structure (i.e., all compounds not introduced as standards), how does one classify the molecular formula? Especially given that one formula could contain many compounds with different structures.*

**Response:**

We appreciate the reviewer's comments, and we have added a detailed explanation in the application section of the quantitative method (Section 3.3). Now it reads as follows:

[revised manuscript text omitted]

**Response:**

Revised accordingly.
* * *
*18. Figure 3. See note for Figure 1 about bar charts with log axes.*

**Response:**

We appreciate the reviewer's comments, and we have remade Figure 3 as a scatter plot in the revised manuscript. Now it shows as follows:

[Figure]

**Figure 4: (a) The sensitivity results of toluene oxidation intermediates obtained at the B3LYP/Def2TZVP (D3) level. (b) The sensitivity of key toluene oxidation intermediates obtained by direct calibration, binding energy semi-quantitative, and voltage scan semi-quantitative methods. All sensitivity values presented in the figures were acquired under the RH condition of approximately 55 ± 5%.**
* * *
*19. Line 342. How are structures assigned for calculating binding enthalpies of unknown products?*

**Response:**

We appreciate the reviewer's comments. In the revised manuscript, we have added a detailed discussion on the structures assigned for calculating the binding enthalpies of unknown toluene oxidation products. Now it reads as follows:

During the photo-oxidation process of precursors such as aromatics, I-CIMS measurements reveal that each formula may have many isomers (Bi et al., 2021a). Therefore, in the semi-quantitative study of toluene oxidation products using a binding energy-based method, it is crucial to reasonably infer their structures. For the oxidation products of toluene, including $C_4H_4O_2$, $C_5H_6O_2$, $C_7H_8O_4$, $C_7H_{10}O_4$, $C_7H_{10}O_5$, and $C_7H_8O_6$, we first excluded furanones or aldehyde species that cannot be measured by I-CIMS among their isomers. Additionally, we excluded isomers originating from lower concentration multi-generation oxidation products. For example, in the toluene system, the $C_7H_8O_4I^-$ signal measured by CIMS reveals three isomers: first-generation products in the bicyclic $RO_2$ pathway, a minor fourth-generation product hydroxyquinol derived from the phenolic pathway, and a second-generation epoxy hydroxy compound from the epoxide pathway. Laboratory experiments have revealed a negligible contribution from the epoxy pathway(Zaytsev et al., 2019), and the impact of second-generation epoxy hydroxy compounds on the $C_7H_8O_4I^-$ signal can be considered negligible. Based on the reasonable inference above, we propose that the signals of $C_4H_4O_2$, $C_5H_6O_2$, $C_7H_8O_4$, $C_7H_{10}O_4$, $C_7H_{10}O_5$, and $C_7H_8O_6$ detected by I-CIMS

primarily originate from the major first-generation products of the bicyclic $RO_2$ pathway as depicted in Figure S9. Due to their diverse functional groups, the multitype functional group species semi-quantitative equations based on the binding energy method is employed for their quantification.

For the multi-generation products $C_7H_8O_3$, $C_4H_4O_3$, and $C_5H_6O_3$, by excluding furanones and aldehydic compounds that are difficult to detect by I-CIMS, it can be inferred that the signal of $C_7H_8O_3$ primarily originates from trihydroxytoluene, the signal of $C_4H_4O_3$ primarily comes from (Z)-4-oxobut-2-enoic acid, and the signal of $C_5H_6O_3$ primarily comes from (Z)-4-oxopent-2-enoic acid and (Z)-2-methyl-4-oxobut-2-enoic acid. $C_7H_8O_3$ quantification involves semi-quantitative equations with polyphenol or diacid species, while for $C_4H_4O_3$ and $C_5H_6O_3$, which are keto acids, semi-quantitative equations incorporating multiple functional group species are used for quantification..

Furthermore, we attempted to employ voltage scanning techniques for the auxiliary identification of isomers. Isaacman et al. preliminarily explored the possible differences in the $dV_{50}$ of isomers (Isaacman-Vanwertz et al., 2018), which may serve as an important means to distinguish and quantify isomers measured by I-CIMS. In the toluene system, the $C_7H_8O$ produced during the reaction could originate from cresol in the phenolic pathway or from benzyl alcohol, a byproduct of the aldehyde pathway. Through voltage scanning, we observed a small difference in the voltage variation of C7H8OI- in the toluene system compared to the cresol standard samples, with dV50 values of -0.97 and -1.12, respectively. This difference may stem from the significantly higher yield of cresol, the primary product in the toluene system, compared to benzyl alcohol (Smith et al., 1998; Baltaretu et al., 2009; Ji et al., 2017), suggesting that the influence of this type of isomerization can be disregarded during the quantification process. Therefore, $C_7H_8O$ quantification is performed using semi-quantitative equations specific to monophenol species. By comparing the voltage scanning results of $C_7H_8O_2I^-$, the oxidation products from the toluene and the dihydroxy toluene sample, and $dV_{50}$ was 0.75 and 0.72, respectively. Therefore, these results indicated that the signal for $C_7H_8O_2I^-$ could be approximated as dihydroxy toluene in the toluene system. Therefore, $C_7H_8O$ quantification is performed using semi-quantitative equations specific to polyphenol or diacid species.
* * ** * *
**Lastly, we would again express our appreciation to the reviewers and editor for their warm-hearted help. Thank you very much!!!!**
* * *

[revised manuscript text omitted]

---

## Author Comment (AC2)

**Response to the Comments of the Reviewers**

Dear Editor and Reviewers,

We would like to thank you and the reviewers for the great efforts and elaborate work on this manuscript.

We revised the manuscript by responding to each of the suggestions in the reviews. In our response, the questions of the reviewers are shown in *Italic* form and the responses in standard form. The corresponding revisions to the manuscript are marked in blue. All updates to the original submission are tracked in the revised manuscript.

We appreciate your help and time.

Sincerely yours,

Xin Li and Co-authors.

College of Environmental Sciences and Engineering
Peking University
100871 Beijing China
E-mail: li_xin@pku.edu.cn
Tel: +86-185 1358 6831
* * *
Manuscript Number: AMT-2024-1203.

Manuscript Title: *Optimizing Iodide-Adduct CIMS Quantitative Method for Toluene Oxidation Intermediates: Experimental Insights into Functional Group Differences.*
* * *
**Response to Reviewer #3**

*General comments*

*This manuscript describes the development, characterization, and application of an optimized semi-quantitative method for toluene oxidation intermediates using iodide-adduct CIMS. The method was established based on the linear correlation between instrument sensitivity to the iodide addition and theoretically calculated binding enthalpies of the formed iodide-analyte adduct for different species categories. Compared to the previous semi-quantitative method based on binding enthalpies of iodide adducts, the categorized semi-quantitative method proposed in this study appears to have a higher quantification accuracy. Although this method was developed particularly for the toluene oxidation products, the concept can be also applied to the development of the quantification method for other oxidation system. This work is scientifically sound and the manuscript is well written. I recommend its publication in AMT after several minor comments are addressed.*

**Response:**

We would like to thank reviewer #3 for carefully reading our manuscript and for the valuable and constructive comments. The manuscript was carefully revised according to the reviewer's suggestions. Listed below are our point-by-point responses to reviewer's comments. In our response, the questions of the reviewers are shown in *Italic* form and the responses in standard form. The corresponding revisions to the manuscript are marked in blue. All updates to the original submission are tracked in the revised manuscript. Lastly, we appreciate the positive feedback from the reviewer.
* * *
*Comment*

*L100: What was the TIC value of the iodide CIMS under typical operating conditions?*

**Response:**

We appreciate the reviewer's comments, and we have added the information of TIC value. Now it reads as follows:

During the experimental operating conditions of this study, the iodide CIMS exhibited a total ion count (TIC) of approximately 2 ions per extraction (ions/ex.) and 32,000 counts per second (cps).
* * *
*L109-110: It would be more informative and readable if the method for the generation of the calibration gas (i.e., using gas cylinders, permeation tubes, or atomizing standard solutions) can be added for each standard in Table S1.*

**Response:**

We appreciate the reviewer's comments, and we have added the information of the calibration method in Table S1 according to the reviewer's suggestions.

In addition, we actually calibrate the I-CIMS sensitivities of 37 organic compounds using certified

penetrant tubes (CPT) and a home-built liquid calibration unit (LCU). Additionally, gas standards from Spectra Gas Inc. were used for the calibration of PTR-QMS to accurately measure nonmethane hydrocarbons (HMHCs), aldehydes (such as benzaldehyde), and other oxidation products that cannot be quantified by I-CIMS during chamber experiments.

Therefore, we have improved the description of this section in the revised manuscript and the SI. Now it reads as follows:

Line 120-124 in tracked changes manuscript: In this study, 37 species with different functional group were directly calibrated using certified penetrant tubes (CPT) and a home-built liquid calibration unit (LCU), including monophenols, monoacids, polyphenols, diacids, phenolic acids, keto acids, furanones, and other species (Table S1). For OVOCs that can be customized to a standard gas and penetrant tubes, calibration is often performed using certified penetrant tubes (KinTek Inc.) at 5-6 gradient concentration levels (Huang et al., 2019).

SI: HMHCs and some oxidation products were calibrated using gas standards (Spectra gas Inc.) and certified penetrant tubes (KinTek Inc.) for PTR-QMS under experimental conditions.

Due to the length of the table, here is a brief display of the changes to the monophenol section of the table:

**Table S1: Detailed information on the direct calibration of species associated with aromatic hydrocarbons and their oxidation products in this study. (Monophenol part)**

| Type | No. | Species | Formula | MW | Detection limit* | Sensitivity (ncps/ppb) | Structure | Methods |
|------|-----|---------|---------|-----|-----------------|------------------------|-----------|---------|
| Monophenol | 1 | m-Cresol | $C_7H_8O$ | 108.06 | 0.083 | $1.3 \times 10^2$ | | CPT |
| | 2 | Phenol | $C_6H_6O$ | 94.04 | 0.108 | $1.5 \times 10^2$ | | CPT |
| | 3 | 2,6-Xylenol | $C_8H_{10}O$ | 122.07 | 4.198 | 2 | | CPT |
| | 4 | Texanol | $C_{12}H_{24}O_3$ | 216.17 | 0.554 | 26 | | LCU |

* The Detection limit unit is ppb in 1-seconds, S/N=3

CPT represent certified penetrant tube, LCU represent home-built liquid calibration unit.
* * *
*L114: Suggest also providing the corresponding RH values at ambient pressure.*

**Response:**

We appreciate the reviewer's comments, and we have added the corresponding RH values at ambient pressure. Now it reads as follows:

To investigate the influence of humidity on calibration, both CPT and LCU calibration system are equipped with a humidification section that can control humidity within the range of 0.12 to 22.00 mmol/mol, which corresponds to a relative humidity (RH) of 0.4% to 70% at a temperature of 25°C and a pressure of 101.325 kPa.
* * *
*L117-119: Did all the standards dissolved into the solvent turn into the gaseous form after atomization? Were any particles detected in the atomized gas flow?*

**Response:**

We are sorry for the unclear expression. In this study, all the standards dissolved in the solvent were mixed with nitrogen of a large flow rate at a microliter-level flow rate to completely evaporate the atomized droplets. No liquid condensation was observed on the wall of the mixing unit, and no particulate matter was present. After sufficient equilibration time, stable signals of standard samples could be detected in I-CIMS. We have carefully revised this statement in the revised manuscript. Now it reads as follows:

The standard sample was mixed with a soluble solvent, including water, dichloromethane, or acetone, and the solvent was atomized at a given flow rate. Subsequently, the atomized gas was mixed with high-flow nitrogen to ensure the complete evaporation of the atomized droplets, which were then injected into the sampling port. No liquid condensation was observed on the wall of the mixing unit, and no particulate matter was present. After sufficient equilibration time, stable signals of standard samples could be detected in I-CIMS.
* * *
*L185: As the sensitivity of iodide ionization is strongly dependent on the RH, the authors should clearly indicate the RH conditions (dry or wet) under which the sensitivity values shown in Figures 1-3 were estimated.*

**Response:**

We appreciate the reviewer's comments and have accordingly included the RH conditions information in the titles of Figures 1-3. The primary humidity note for the titles of Figures 1-4 is as follows:

All sensitivity values presented in the figures were acquired under the RH condition of approximately 55 ± 5%.
* * *
*L307-308: Figure S8 provides a nice validation of the proposed semi-quantitative method, I would suggest moving the figure to the main text.*

**Response:**

We appreciate the reviewer's comments, and we have moved Figure S8 into the main text and incorporated additional validation analyses for the semi-quantitative method. The revised figure is presented as follows:

[Figure]

**Figure 3: (a)** The difference between the measured sensitivity and the calculated sensitivity for standards at the B3LYP/Def2TZVP (D3) level. **(b)** The uncertainty of classification-based semi-quantitative methods at the B3LYP/Def2TZVP (D3) level. The uncertainty is computed by dividing the absolute difference between the measured sensitivity and the calculated sensitivity by the measured sensitivity. All sensitivity values presented in the figures were acquired under the RH condition of approximately $55 \pm 5\%$.
* * *
*L309: "with absolute deviations below 40%". This value is actually the relative deviation.*

**Response:**

"absolute deviations" has been replaced by "relative deviations"
* * *
*L312-313: How were these uncertainties quantified?*

**Response:**

We appreciate the reviewer's comments. In the revised paper, we have included a detailed analysis of uncertainty quantification. Now it reads as follows:

Based on the above discussion, we use the interquartile range (IQR) from the uncertainty box plot in Figure 3b to evaluate the classification-based semi-quantitative method. For species with standards, the uncertainties in sensitivity are approximately 25%-50%. For species without standards, semi-quantitative sensitivity uncertainty may increase due to improper classification. This is represented by the IQR of the error classification uncertainty prediction box plot (C-ECP in Figure 3b), ranging from 25% to 80%. Additionally, our previous studies have shown that mass spectrometric sampling losses can introduce uncertainties of approximately 10%-20% in the measurement of oxidation intermediates (Huang et al., 2019). In this study, the overall uncertainty for oxidation intermediates ranged from 30% to 85%, which is calculated as the quadrature addition of individual uncertainties.
* * *
*L338-339: There may be isomers for the oxidation products identified here. Did the sensitivity values estimated here take into account the sensitivity differences between potential isomers?*

**Response:**

We appreciate the reviewer's comments. Isomers pose a significant challenge in mass spectrometry data analysis. We have included a discussion on isomers in the revised manuscript. Now it reads as follows:

During the photo-oxidation process of precursors such as aromatics, I-CIMS measurements reveal that each formula may have many isomers (Bi et al., 2021). Therefore, in the semi-quantitative study of toluene oxidation products using a binding energy-based method, it is crucial to reasonably infer their structures. For the oxidation products of toluene, including $C_4H_4O_2$, $C_5H_6O_2$, $C_7H_8O_4$, $C_7H_{10}O_4$, $C_7H_{10}O_5$, and $C_7H_8O_6$, we first excluded furanones or aldehyde species that cannot be measured by I-CIMS among their isomers. Additionally, we excluded isomers originating from lower concentration multi-generation oxidation products. For example, in the toluene system, the $C_7H_8O_4I^-$ signal measured by CIMS reveals three isomers: first-generation products in the bicyclic $RO_2$ pathway, a minor fourth-generation product hydroxyquinol derived from the phenolic pathway, and a second-generation epoxy hydroxy compound from the epoxide pathway. Laboratory experiments have revealed a negligible contribution from the epoxy pathway(Zaytsev et al., 2019), and the impact of second-generation epoxy hydroxy compounds on the $C_7H_8O_4I^-$ signal can be considered negligible. Based on the reasonable inference above, we propose that the signals of $C_4H_4O_2$, $C_5H_6O_2$, $C_7H_8O_4$, $C_7H_{10}O_4$, $C_7H_{10}O_5$, and $C_7H_8O_6$ detected by I-CIMS primarily originate from the major first-generation products of the bicyclic $RO_2$ pathway as depicted in Figure S9. Due to their diverse functional groups, the multitype functional group species semi-quantitative equations based on the binding energy method is employed for their quantification.

For the multi-generation products $C_7H_8O_3$, $C_4H_4O_3$, and $C_5H_6O_3$, by excluding furanones and aldehydic compounds that are difficult to detect by I-CIMS, it can be inferred that the signal of $C_7H_8O_3$ primarily originates from trihydroxytoluene, the signal of $C_4H_4O_3$ primarily comes from (Z)-4-oxobut-2-enoic acid, and the signal of $C_5H_6O_3$ primarily comes from (Z)-4-oxopent-2-enoic acid and (Z)-2-methyl-4-oxobut-2-enoic acid. $C_7H_8O_3$ quantification involves semi-quantitative equations with polyphenol or diacid species, while for $C_4H_4O_3$ and $C_5H_6O_3$, which are keto acids, semi-quantitative equations incorporating multiple functional group species are used for quantification..

Furthermore, we attempted to employ voltage scanning techniques for the auxiliary identification of isomers. Isaacman et al. preliminarily explored the possible differences in the $dV_{50}$ of isomers (Isaacman-Vanwertz et al., 2018), which may serve as an important means to distinguish and quantify isomers measured by I-CIMS. In the toluene system, the $C_7H_8O$ produced during the reaction could originate from cresol in the phenolic pathway or from benzyl alcohol, a byproduct of the aldehyde pathway. Through voltage scanning, we observed a small difference in the voltage variation of C7H8OI- in the toluene system compared to the cresol standard samples, with dV50 values of -0.97 and -1.12, respectively. This difference may stem from the significantly higher yield of cresol, the primary product in the toluene system, compared to benzyl alcohol (Smith et al., 1998; Baltaretu et al., 2009; Ji et al., 2017), suggesting that the influence of this type of isomerization can be disregarded during the quantification process. Therefore, $C_7H_8O$ quantification is performed using semi-quantitative equations specific to monophenol species. By comparing the voltage scanning results of $C_7H_8O_2I^-$, the oxidation products from the toluene and the dihydroxy toluene sample, and $dV_{50}$ was 0.75 and 0.72, respectively. Therefore, these results indicated that the signal for $C_7H_8O_2I^-$ could be approximated as dihydroxy toluene in the toluene system. Therefore, $C_7H_8O$ quantification is performed using semi-quantitative equations specific to polyphenol or diacid species.
* * *
*L349: Replace "less" by "small". Also, "absolute deviations" should be "relative deviations".*

**Response:**

Revised accordingly.
* * *
*L357: How was the "satisfactory" defined or determined here? 4*

**Response:**

We are sorry for the confusing expression. We have carefully revised this statement in the revised manuscript. Now it reads as follows:

In this study, the fitting of the experimental and simulated peak shapes exhibited a high degree of correspondence ($R^2 > 0.99$), indicating that fitting species' molecular ion peak areas using high-resolution mass spectrometry can represent the measured species signal effectively.
* * *
*L360: Were the concentrations of toluene oxidation products corrected for the RH effect?*

**Response:**

We appreciate the reviewer's comments. When establishing the semi-quantitative equation, we accounted for the impact of humidity. However, during the chamber experiments, since the humidity was consistently maintained at approximately $55 \pm 5\%$, which coincides with the humidity condition specified for the semi-quantitative equation depicted in Figure 3, the correction factor ($RH_{corr}$) in the semi-quantitative process can be designated as 1. We have carefully revised this statement in the revised manuscript. Now it reads as follows:

During the chamber experiment, the humidity was maintained at approximately $55 \pm 5\%$, which is consistent with the humidity condition established for the semi-quantitative equation in Figure 3, the correction factor ($RH_{corr}$) in the semi-quantitative process can be set to 1. Time series of the oxidation products in the toluene + OH system at different concentrations of NO is shown in Figure 4 and Figure S11.
* * *
*L371-372: References should be provided for this statement.*

**Response:**

We appreciate the reviewer's comments, and we have the reference in the revised manuscript. Now it reads as follows:

These compounds were recently identified as ring-opening products originating from the 1,5-aldehydic H-shift of alkoxy radicals during the bicyclic peroxy radical pathway (Xu et al., 2020).
* * ** * *
**Lastly, we would again express our appreciation to the reviewers and editor for their warm-hearted help. Thank you very much!!!!**
* * *
**References**

Baltaretu, C. O., Lichtman, E. I., Hadler, A. B., and Elrod, M. J.: Primary Atmospheric Oxidation Mechanism for Toluene, J. Phys. Chem. A., 113, 221-230, https://doi.org/10.1021/jp806841t, 2009.

Bi, C., Krechmer, J. E., Frazier, G. O., Xu, W., Lambe, A. T., Claflin, M. S., Lerner, B. M., Jayne, J. T., Worsnop, D. R., Canagaratna, M. R., and Isaacman-VanWertz, G.: Coupling a gas chromatograph simultaneously to a flame ionization detector and chemical ionization mass spectrometer for isomer-resolved measurements of particle-phase organic compounds, Atmos. Meas. Tech., 14, 3895-3907, https://doi.org/10.5194/amt-14-3895-2021, 2021.

Huang, G., Liu, Y., Shao, M., Li, Y., Chen, Q., Zheng, Y., Wu, Z., Liu, Y., Wu, Y., Hu, M., Li, X., Lu, S., Wang, C., Liu, J., Zheng, M., and Zhu, T.: Potentially Important Contribution of Gas-Phase Oxidation of Naphthalene and Methylnaphthalene to Secondary Organic Aerosol during Haze Events in Beijing, Environ. Sci. Technol., 53, 1235-1244, https://doi.org/10.1021/acs.est.8b04523, 2019.

Isaacman-VanWertz, G., Massoli, P., O'Brien, R., Lim, C., Franklin, J. P., Moss, J. A., Hunter, J. F., Nowak, J. B., Canagaratna, M. R., Misztal, P. K., Arata, C., Roscioli, J. R., Herndon, S. T., Onasch, T. B., Lambe, A. T., Jayne, J. T., Su, L., Knopf, D. A., Goldstein, A. H., Worsnop, D. R., and Kroll, J. H.: Chemical evolution of atmospheric organic carbon over multiple generations of oxidation, Nat. Chem., 10, 462-468, https://doi.org/10.1038/s41557-018-0002-2, 2018.

Ji, Y., Zhao, J., Terazono, H., Misawa, K., Levitt, N. P., Li, Y., Lin, Y., Peng, J., Wang, Y., and Duan, L.: Reassessing the atmospheric oxidation mechanism of toluene, Proc. Natl. Acad. Sci. USA., 114, 8169-8174, https://doi.org/10.1073/pnas.1705463114, 2017.

Smith, D. F., McIver, C. D., and Kleindienst, T. E.: Primary product distribution from the reaction of hydroxyl radicals with toluene at ppb NOX mixing ratios, J. Atmos. Chem., 30, 209-228, 10.1023/a:1005980301720, 1998.

Zaytsev, A., Koss, A. R., Breitenlechner, M., Krechmer, J. E., Nihill, K. J., Lim, C. Y., Rowe, J. C., Cox, J. L., Moss, J., Roscioli, J. R., Canagaratna, M. R., Worsnop, D. R., Kroll, J. H., and Keutsch, F. N.: Mechanistic study of the formation of ring-retaining and ring-opening products from the oxidation of aromatic compounds under urban atmospheric conditions, Atmos. Chem. Phys., 19, 15117-15129, https://doi.org/10.5194/acp-19-15117-2019, 2019.

---

## Author Comment (AC3)

**Response to the Comments of the Reviewers**

Dear Editor and Reviewers,

We would like to thank you and the reviewers for the great efforts and elaborate work on this manuscript.

We revised the manuscript by responding to each of the suggestions in the reviews. In our response, the questions of the reviewers are shown in *Italic* form and the responses in standard form. The corresponding revisions to the manuscript are marked in blue. All updates to the original submission are tracked in the revised manuscript.

We appreciate your help and time.

Sincerely yours,

Xin Li and Co-authors.

College of Environmental Sciences and Engineering
Peking University
100871 Beijing China
E-mail: li_xin@pku.edu.cn
Tel: +86-185 1358 6831
* * *
*Manuscript Number: AMT-2024-1203.*

*Manuscript Title: Optimizing Iodide-Adduct CIMS Quantitative Method for Toluene Oxidation Intermediates: Experimental Insights into Functional Group Differences.*
* * *
**Response to Reviewer #1**

*General comments*

*In this manuscript, the authors present an attempt to quantify toluene oxidation intermediates by establishing quantitative and semi-quantitative calibration methods for I-CIMS. Specifically, the authors established semi-quantitative equations for four distinct categories: monophenols, monoacids, polyphenol or diacid species, and species with multiple functional groups. This classification method enhances the accuracy of the semi-quantitative approach ($R^2$ from ~0.50 to >0.88). Overall, the research goal of this manuscript is novel and has practical atmospheric significance. The description of the calibration methods and experimental results is logical and comprehensive. After the authors address the minor comments below, the manuscript can be published in AMT.*

**Response:**

We would like to thank reviewer #1 for carefully reading our manuscript and for your valuable and constructive comments. The manuscript was carefully revised according to the reviewer's suggestions. Listed below are our point-by-point responses to reviewer's comments. In our response, the questions of the reviewers are shown in *Italic* form and the responses in standard form. The corresponding revisions to the manuscript are marked in blue. All updates to the original submission are tracked in the revised manuscript. Lastly, we would like to thank reviewer for the positive comments again.
* * *
*Comments*

*1. Line 42, "H3O+ ions are used for the detection of VOCs". This description here is not accurate. Conventional PTR has the ability to detect some I/SVOCs, although not many species. The newly developed Vocus or Fusion PTR can detect more I/SVOCs, some of which are oxygenated compounds (e.g., Atmos. Meas. Tech. 2019, 2403-2421).*

**Response:**

We are sorry for the inaccurate expression. We have carefully revised this statement in the revised manuscript. Now it reads as follows:

Among them, $H_3O^+$ ions in conventional PTR instrument are designed to primarily measure VOCs (Yuan et al., 2017; Riva et al., 2019). However, with the inlet modifications introduced in the newly developed VOCUS or FUSION PTR instruments, the $H_3O^+$ ions were able to measure a large range of OVOCs (Reinecke et al., 2023; Riva et al., 2019). Moreover, $NH_4^+$ ions are used for the detection of oxygenated organic compounds, including alcohols, aldehydes, ketones (Hansel et al., 2018; Xu et al., 2022), $NO_3^-$ ions are used for the detection of highly oxygenated organic molecules (HOMs) (Xu et al., 2020), $I^-$ ions are used for the detection of compounds containing many different functional groups, including monophenols, polyphenols, monoacids, diacids, phenolic acids, keto acids, and inorganic species (Lee et al., 2014), $CH_3CH(O)O^-$ ions are used for the detection of

organic acids (Hansel et al., 2018), and $CF_3O^-$ ions are used for the detection of oxygenated organics, including hydroperoxides (Schwantes et al., 2017).
* * *
*2. Lines 53-76, some key references are missing when introducing the calibration method of CIMS and its research progress. For example, Li et al. (Environ. Sci. Technol. 2021, 12841-12851) used 22 organic standards to calibrate I-CIMS and reduced the uncertainty in total organic carbon concentrations to ~20%-35% when combining the voltage scanning approach.*

**Response:**

We appreciate the reviewer's comments, and we have added some literature introductions of CIMS calibration methods in the revised manuscript. Now it reads as follows:

First, the quantification of oxidation intermediates requires calibration using commercial standards. Direct calibration plays a crucial role in exploring the sensitivity characteristics of instruments and reducing the uncertainty of quantitative methods. Common direct calibration methods include the utilization of standard gas cylinders (SGC), penetrant tubes (CPT), and liquid calibration units (LCU) (Xu et al., 2022; Huang et al., 2019). Xu et al. employed 60 organic compound standards utilizing SGC and a home-built LCU method to calibrate $NH_4^+$ CIMS, revealing its differential sensitivity to diverse organic compounds (Xu et al., 2022). Additionally, Li et al. implemented 22 organic standards with the LCU method for I-CIMS calibration, achieving a significant reduction in total organic carbon concentration uncertainty to approximately 20%-35% when coupled with the voltage scanning approach (Li et al., 2021). However, most intermediates measurable by I-CIMS are difficult to synthesize effectively as pure standards.
* * *
*3. Line 172, "humidity" should be "relative humidity".*

**Response:**

Revised accordingly.
* * *
*4. Section 3.1, it would be more straightforward to list the sensitivities in a table, probably in Table S1.*

**Response:**

We appreciate the reviewer's comments, and we have listed the sensitivities in Table S1 in the revised manuscript.
* * *
*5. Line 230, the sections and figures in the SI should be presented in order in the manuscript.*

**Response:**

We appreciate the reviewer's comments, and we have combined the sections related to humidity, thereby ensuring the correct order of presentation for sections and figures. Now the sentence reads as follows:

Through the establishment of humidity-dependent parametric equations, species sensitivity under different humidity conditions can be obtained, as detailed in Section S3, Figure S3 and Figure S4. These humidity-dependent parametric equations correspond to $RH_{Corr}$ in Equation (2), (4), and (S2).

Additionally, we have meticulously reviewed the sequence of sections and figures throughout the SI file, ensuring they were presented in order in the manuscript.
* * *
*6. Figure 2, it is difficult to match the data points to compound names. Adding some arrows may help.*

**Response:**

We appreciate the reviewer's comments, and we have added some arrows in the Figure 2 to match the data points to compound names. The revised Figure 2 is depicted as follows:

[Figure]

**Figure 2: Fitting curve for cluster binding enthalpies and logarithmic sensitivities of (a) monophenol species (b) monoacid species, (c) polyphenol or diacid species, and (d) multitype functional group species at the B3LYP/Def2TZVP (D3) level. All sensitivity values presented in the figures were acquired under the RH condition of approximately 55 ± 5%.**
* * *
*7. Lines 363-370, could the authors show the time series of C7H8O4, C7H10O4, C7H10O5, C4H4O3, and C5H6O3 separately somewhere in the SI? It would be helpful to see the ratio of C5H6O3 to C4H4O3 as well.*

**Response:**

We appreciate the reviewer's comments, and we have added the time series of $C_7H_8O_4$, $C_7H_{10}O_4$, $C_7H_{10}O_5$, $C_4H_4O_3$, and $C_5H_6O_3$ separately in Figure S11. Now it shows as follows:

[Figure]

**Figure S11: Time series of (a) glyoxal, (b) methyl glyoxal, (c) C$_4$H$_4$O$_3$, (d) C$_5$H$_6$O$_3$, (e) C$_7$H$_8$O$_4$, (g) C$_7$H$_{10}$O$_4$, and (g) C$_7$H$_{10}$O$_5$ during the oxidation of toluene without NO injection (blue) and with NO injection (orange). Note: glyoxal and methyl glyoxal were measured by CEAS instrument.**

In addition, we calculated the ratio of C$_5$H$_6$O$_3$ to C$_4$H$_4$O$_3$, which were 3.77 and 1.70 under low and high NO conditions, respectively. Now it reads as follows:

Additionally, the study found that the ratio of C$_5$H$_6$O$_3$ to C$_4$H$_4$O$_3$ under low NO and high NO conditions was 3.77 and 1.70, respectively. These compounds were recently identified as ring-opening products originating from the 1,5-aldehydic H-shift of alkoxy radicals during the bicyclic peroxy radical pathway. The ratio of C$_5$H$_6$O$_3$ to C$_4$H$_4$O$_3$ exhibited an inverse relationship with NOx levels, similar to the case of the traditional ring-opening products C$_5$H$_6$O$_2$ to C$_4$H$_4$O$_2$ (He et al., 2023). The assumption of fixed ratios of ring-opening products in previous studies may have been due to a lack of measurement and quantification of oxidized intermediates.
* * *
*8. Lines 401-403, there is no need to repeat these numbers in the conclusion.*

**Response:**

We appreciate the reviewer's comments, and we have removed the duplicated sentences regarding the ratios described in the conclusion (Lines 401-403) in the revised manuscript.
* * *
*9. Check the capitalization of the first letter. Some examples: Line 60, "Discovered"; Line 149, "Computational"; Line 213, "2,6-Xylenol and Texanol"; Compound names in Figure 2; Line 288, "Salicylic acid, Citric acid".*

**Response:**

We appreciate the reviewer's comments. We have carefully checked the capitalization of the first letter in the manuscript and carefully revised according to the reviewer's suggestions.
* * *
**Lastly, we would again express our appreciation to the reviewers and editor for their warm-hearted help. Thank you very much!!!!**
* * *
**References**

Hansel, A., Scholz, W., Mentler, B., Fischer, L., and Berndt, T.: Detection of RO2 radicals and other products from cyclohexene ozonolysis with NH4+ and acetate chemical ionization mass spectrometry, Atmos. Environ., 186, 248-255, https://doi.org/10.1016/j.atmosenv.2018.04.023, 2018.

He, S., Liu, Y., Song, M., Li, X., Lu, S., Chen, T., Mu, Y., Lou, S., Shi, X., Qiu, X., Zhu, T., and Zhang, Y.: Insights into the Peroxide-Bicyclic Intermediate Pathway of Aromatic Photooxidation: Experimental Yields and NOx-Dependency of Ring-Opening and Ring-Retaining Products, Environ. Sci. Technol., 57, 20657-20668, https://doi.org/10.1021/acs.est.3c05304, 2023.

Huang, G., Liu, Y., Shao, M., Li, Y., Chen, Q., Zheng, Y., Wu, Z., Liu, Y., Wu, Y., Hu, M., Li, X., Lu, S., Wang, C., Liu, J., Zheng, M., and Zhu, T.: Potentially Important Contribution of Gas-Phase Oxidation of Naphthalene and Methylnaphthalene to Secondary Organic Aerosol during Haze Events in Beijing, Environ. Sci. Technol., 53, 1235-1244, https://doi.org/10.1021/acs.est.8b04523, 2019.

Lee, B. H., Lopez-Hilfiker, F. D., Mohr, C., Kurten, T., Worsnop, D. R., and Thornton, J. A.: An Iodide-Adduct High-Resolution Time-of-Flight Chemical-Ionization Mass Spectrometer: Application to Atmospheric Inorganic and Organic Compounds, Environ. Sci. Technol., 48, 6309-6317, https://doi.org/10.1021/es500362a, 2014.

Li, K., Wentzell, J. J. B., Liu, Q., Leithead, A., Moussa, S. G., Wheeler, M. J., Han, C., Lee, P., Li, S.-M., and Liggio, J.: Evolution of Atmospheric Total Organic Carbon from Petrochemical Mixtures, Environ. Sci. Technol., https://doi.org/10.1021/acs.est.1c02620, 2021.

Reinecke, T., Leiminger, M., Jordan, A., Wisthaler, A., and Müller, M.: Ultrahigh Sensitivity PTR-MS Instrument with a Well-Defined Ion Chemistry, Anal. Chem., 95, 11879-11884, https://doi.org/10.1021/acs.analchem.3c02669, 2023.

Riva, M., Rantala, P., Krechmer, J. E., Peräkylä, O., Zhang, Y., Heikkinen, L., Garmash, O., Yan, C., Kulmala, M., Worsnop, D., and Ehn, M.: Evaluating the performance of five different chemical ionization techniques for detecting gaseous oxygenated organic species, Atmos. Meas. Tech., 12, 2403-2421, https://doi.org/10.5194/amt-12-2403-2019, 2019.

Schwantes, R. H., Schilling, K. A., McVay, R. C., Lignell, H., Coggon, M. M., Zhang, X., Wennberg, P. O., and Seinfeld, J. H.: Formation of highly oxygenated low-volatility products from cresol oxidation, Atmos. Chem. Phys., 17, 3453-3474, https://doi.org/10.5194/acp-17-3453-2017, 2017.

Xu, L., Moller, K. H., Crounse, J. D., Kjaergaard, H. G., and Wennberg, P. O.: New Insights into the Radical Chemistry and Product Distribution in the OH-Initiated Oxidation of Benzene, Environ. Sci. Technol., 54, 13467-13477, https://doi.org/10.1021/acs.est.0c04780, 2020.

Xu, L., Coggon, M. M., Stockwell, C. E., Gilman, J. B., Robinson, M. A., Breitenlechner, M., Lamplugh, A., Crounse, J. D., Wennberg, P. O., Neuman, J. A., Novak, G. A., Veres, P. R., Brown, S. S., and Warneke, C.: Chemical ionization mass spectrometry utilizing ammonium ions (NH4+ CIMS) for measurements of organic compounds in the atmosphere, Atmos. Meas. Tech., 15, 7353-7373, https://doi.org/10.5194/amt-15-7353-2022, 2022.

Yuan, B., Koss, A. R., Warneke, C., Coggon, M., Sekimoto, K., and de Gouw, J. A.: Proton-Transfer-Reaction Mass Spectrometry: Applications in Atmospheric Sciences, Chem. Rev., 117, 13187-13229, https://doi.org/10.1021/acs.chemrev.7b00325, 2017.